# MoLEx: Mixture of Layer Experts for Fine-tuning with Sparse Upcycling

**Rachel S.Y. Teo**
Department of Mathematics
National University of Singapore
rachel.tsy@u.nus.edu

**Tan M. Nguyen**
Department of Mathematics
National University of Singapore
tanmn@nus.edu.sg

## Abstract

Large-scale pre-training of deep models, followed by fine-tuning them to adapt to downstream tasks, has become the cornerstone of natural language processing (NLP). The prevalence of vast corpses of data coupled with computational resources has led to large models with a considerable number of parameters. While the massive size of these models has led to remarkable success in many NLP tasks, a detriment is the expense required to retrain all the base model's parameters for the adaptation to each task or domain. Parameter Efficient Fine-Tuning (PEFT) provides a highly effective solution for this challenge by minimizing the number of parameters required to be trained in adjusting to the new task while maintaining the quality of the model. While existing methods have achieved impressive results, they mainly focus on adapting a subset of parameters using adapters, weight reparameterization, and prompt engineering. In this paper, we study layers as extractors of different types of linguistic information that are valuable when used in conjunction with each other. We then propose the Mixture of Layer Experts (MoLEx), a novel sparse mixture of experts (SMoE) whose experts are layers in the pre-trained model. In particular, MoLEx is applied at each layer of the pre-trained model. It performs a conditional computation of a mixture of layers during fine-tuning to provide the model with more structural knowledge about the data. By providing an avenue for information exchange between layers, MoLEx enables the model to make a more well-informed prediction for the downstream task, leading to better fine-tuning results with the same number of effective parameters. As experts can be processed in parallel, MoLEx introduces minimal additional computational overhead. We empirically corroborate the advantages of MoLEx when combined with popular PEFT baseline methods on a variety of downstream fine-tuning tasks, including the popular GLUE benchmark for natural language understanding (NLU) as well as the natural language generation (NLG) End-to-End Challenge (E2E). The code is publicly available at https://github.com/rachtsy/molex.

## 1 Introduction

Numerous natural language processing (NLP) applications depend on leveraging a large-scale, pre-trained language model for multiple downstream tasks (Liu, 2020; Zhu et al., 2020; Stickland et al., 2020; Zhang et al., 2020; Raffel et al., 2020a; Kale & Rastogi, 2020; Zhong et al., 2020; Liu & Lapata, 2019). This adaptation is typically achieved through fine-tuning, a process that involves updating all the parameters of the pre-trained model. Although fine-tuning large language models (LLMs) has driven impressive success across various NLP tasks (Devlin et al., 2018; Liu, 2019; Radford et al., 2019; Raffel et al., 2020b), a drawback is the high computational cost associated with retraining all of the base model's parameters for adaptation to each specific task or domain (Brown et al., 2020; Chowdhery et al., 2023). Parameter efficient fine-tuning (PEFT), such as Low-Rank Adaptation (LoRA) (Hu et al., 2021), offers an effective solution to this issue by reducing the number of parameters that need to be trained for task adaptation while still preserving the model's performance (Zaken et al., 2021; Rücklé et al., 2021; Xu et al., 2023; Pfeiffer et al., 2021; Lin et al.,

---

Correspondence to: rachel.tsy@u.nus.edu and tanmn@nus.edu.sg

2020; Houlsby et al., 2019; Li & Liang, 2021; Xu et al., 2023). Since scaling up language models has proven highly successful, extending this scalability to the fine-tuning process is a desirable goal. However, achieving scalable fine-tuning with parameter efficiency remains a challenging and unresolved problem.

Recently, Sparse Mixture of Experts (SMoE) has emerged as a promising approach to the efficient scaling of language models (Shazeer et al., 2017; Fedus et al., 2022). By dividing the network into modular components and activating only a subset of experts for each input, SMoE retains constant computational costs while enhancing model complexity. This technique has enabled the development of billion-parameter models and has achieved notable success in diverse areas such as machine translation (Lepikhin et al., 2021), image classification (Riquelme et al., 2021), and speech recognition (Kumatani et al., 2021).

## 1.1 SPARSE MIXTURE OF EXPERTS

An MoE replaces a component in the layer of the model, for example, a feed-forward or convolutional layer, by a set of networks termed experts. This approach largely scales up the model but increases the computational cost. An SMoE inherits the extended model capacity from MoE but preserves the computational overhead by taking advantage of conditional computation. In particular, a SMoE consists of a router and $E$ expert networks, $u_i, i = 1, 2, \ldots, E$. For each input token $\boldsymbol{x}_t \in \mathbb{R}^D$ at layer $t$, the SMoE's router computes the affinity scores between $\boldsymbol{x}_t$ and each expert as $g_i(\boldsymbol{x}_t)$, $i = 1, 2, \ldots, E$. In practice, we often choose the router $\boldsymbol{g}(\boldsymbol{x}_t) = [g_1(\boldsymbol{x}_t), g_2(\boldsymbol{x}_t), \ldots, g_E(\boldsymbol{x}_t)]^\top = \boldsymbol{W}\boldsymbol{x} + \boldsymbol{b}$, where $\boldsymbol{W} \in \mathbb{R}^{E \times D}$ and $\boldsymbol{b} \in \mathbb{R}^E$. Then, a sparse gating function TopK is applied to select only $K$ experts with the greatest affinity scores. Here, we define the TopK function as:

$$\mathrm{TopK}(g_i) := \begin{cases} g_i, & \text{if } g_i \text{ is in the } K \text{ largest elements of } g \\ -\infty, & \text{otherwise.} \end{cases} \tag{1}$$

The outputs from $K$ expert networks chosen by the router are then linearly combined as

$$\boldsymbol{x}_{t+1} = \boldsymbol{x}_t + \sum_{i=1}^{E} \mathrm{softmax}(\mathrm{TopK}(g_i(\boldsymbol{x}_t))u_i(\boldsymbol{x}_t) = \boldsymbol{x}_t + u(\boldsymbol{x}_t), \tag{2}$$

where $\mathrm{softmax}(g_i) := \exp(g_i)/\sum_{j=1}^{E} \exp(g_j)$. We often set $K = 2$, i.e., top-2 routing, as this configuration has been shown to provide the best trade-off between training efficiency and testing performance (Lepikhin et al., 2021; Du et al., 2022; Zhou et al., 2023c; Nielsen et al., 2025; Nguyen et al., 2025).

**Sparse Upcycling.** Sparse upcycling (Komatsuzaki et al., 2022) is used to turn a dense pre-trained model into an SMoE model by replacing some multilayer perceptron layers (MLP) in the pre-trained model by SMoE layers. Each SMoE layer contains a fixed number of experts. Each expert is initialized as a copy of the original MLP.

## 1.2 CONTRIBUTION

In this paper, we integrate SMoE into the parameter efficient fine-tuning of large language models. Given a dense pre-trained model, we employ sparse upcycling (Komatsuzaki et al., 2022) to upgrade the model to an SMoE, whose experts are layers in the pre-trained models, and propose the novel Mixture of Layer Experts (MoLEx) upcycling method. MoLEx operates on every layer of the pre-trained model, implementing a conditional computation mechanism that aggregates multiple layers during the fine-tuning process. This approach enriches the model's structural understanding of the data. By facilitating inter-layer information exchange, MoLEx enhances the model's ability to make more informed predictions on downstream tasks, resulting in improved fine-tuning outcomes without increasing the effective parameter count. Furthermore, the parallel processing capability of experts in MoLEx ensures that the additional computational burden is negligible. In summary, our contribution is three-fold.

1. We develop the Mixture of Layer Experts (MoLEx), a new layer-wise sparse upcycling method for the parameter-efficient fine-tuning of LLMs whose experts are layers in the pre-trained model.

2. We study MoLEx from an ensemble model perspective and theoretically prove that a linear MoLEx-upcycled model is more robust than the original dense model.

3. We conduct a layer probe analysis at each MoLEx layer to gain insights into which relevant linguistic information is captured by selected experts for various tasks.

We empirically demonstrate the advantages of MoLEx in accuracy, robustness, and zero-shot transfer learning ability on various large-scale fine-tuning benchmarks, including GLUE (Wang et al., 2018) and the E2E NLG Challenge (Novikova et al., 2017b).

## 2 MoLEx: Mixture of Layer Experts

### 2.1 Backbone Architecture Setting

Our proposed method, MoLEx, is agnostic to the training objective, so it can be adapted to any type of backbone architecture. Without loss of generality and for the convenience of presenting our method, we focus on language modeling as our motivating use case. We first provide a setting for the backbone architecture. Given an input sequence $\boldsymbol{x} \in \mathcal{X}$, where $\mathcal{X} = \mathbb{R}^{N \times D_x}$, we consider the backbone architecture to be a deep model $f$ that transforms the input data point $\boldsymbol{x}$ into its features $\boldsymbol{z}_T \in \mathcal{Z}$, where $\mathcal{Z} = \mathbb{R}^{N \times D_z}$, via a sequence of $T$ processing layers $(u_0, u_1, \ldots, u_{T-1})$ as follows:

$$\boldsymbol{z}_0 = \boldsymbol{x}; \ \boldsymbol{z}_{t+1} = \boldsymbol{z}_t + u_t(\boldsymbol{z}_t; \boldsymbol{\theta}_t), \ t = 0, \ldots, T-1. \tag{3}$$

where $\boldsymbol{\theta}_t$ is the learnable parameters of the processing layer $t$.

**Fine-tuning:** Given a backbone architecture initialized at the learned parameters from the pre-training, fine-tuning is to adapt this model to a downstream task represented by a training dataset of context-target pairs: $\mathcal{Z} = \{(\boldsymbol{x}_i, \boldsymbol{y}_i)\}_{i=1,\ldots,N}$, where both $\boldsymbol{x}_i$ and $\boldsymbol{y}_i$ are sequence of tokens. During full fine-tuning, the model is initialized to pre-trained weights $\Theta^{(0)} = \{\boldsymbol{\theta}_0^{(0)}, \boldsymbol{\theta}_1^{(0)}, \ldots, \boldsymbol{\theta}_{T-1}^{(0)}\}$ and updated to $\Theta^{(0)} + \Delta\Theta = \{\boldsymbol{\theta}_0^{(0)} + \Delta\boldsymbol{\theta}_0, \boldsymbol{\theta}_1^{(0)} + \Delta\boldsymbol{\theta}_1, \ldots, \boldsymbol{\theta}_{T-1}^{(0)} + \Delta\boldsymbol{\theta}_{T-1}\}$ by repeatedly following the gradient to maximize the conditional language modeling objective: $\max_{\Theta} \sum_{(\boldsymbol{x},\boldsymbol{y}) \in \mathcal{Z}} \sum_{j=1}^{|\boldsymbol{y}|} \log(P_{\Theta}(\boldsymbol{y}_j | \boldsymbol{x}, \boldsymbol{y}_{<j}))$.

### 2.2 MoLEx Upcycling

Given the same setting as in Section 2.1, the MoLEx transform is applied on each layer $t$ of the pre-trained model $f^{(0)}$ to turn $f^{(0)}$ into a sparsely upcycled model $\text{MoLEx}(f^{(0)})$ as follows:

$$\boldsymbol{z}_0 = \boldsymbol{x}, \ v_t(\boldsymbol{z}_t) = \sum_{j=0}^{T-1} \text{softmax}(\text{TopK}(g_j(\boldsymbol{z}_t)) u_j(\boldsymbol{z}_t; \boldsymbol{\theta}_j^{(0)}), \ t = 0, \ldots, T-1, \tag{4}$$

$$\boldsymbol{z}_{t+1} = \boldsymbol{z}_t + \alpha u_t(\boldsymbol{z}_t; \boldsymbol{\theta}_t^{(0)}) + (1-\alpha)v_t(\boldsymbol{z}_t),$$

where, again, the sparse gating function TopK selects the top-$K$ layers with highest affinity scores $g_j, j = 0, \ldots, T-1$, where $K$ is set to 1 in our method, and $\text{softmax}(g_i) := \exp(g_i)/\sum_{j=0}^{T-1} \exp(g_j)$ is the softmax normalization operator as defined in Section 1.1. We follow the standard setting for SMoE in (Shazeer et al., 2017; Fedus et al., 2022; Teo & Nguyen, 2024) and choose the router $\boldsymbol{g}(\boldsymbol{z}_t) = [g_0(\boldsymbol{z}_t), g_1(\boldsymbol{z}_t), \ldots, g_{T-1}(\boldsymbol{z}_t)]^\top = \boldsymbol{W}\boldsymbol{z}_t + \boldsymbol{b}$, where $\boldsymbol{W} \in \mathbb{R}^{T \times D_z}$ and $\boldsymbol{b} \in \mathbb{R}^T$. Finally, $\alpha$ is a learnable parameter used to combine the original layer $u_t$ with the chosen layer $v_t$ from the SMoE, $t = 0, \ldots, T-1$. Compared to the original pre-trained model $f^{(0)}$, the MoLEx upcycling $\text{MoLEx}(f^{(0)})$ shares the layer parameters $\Theta = \{\boldsymbol{\theta}_0, \boldsymbol{\theta}_1, \ldots, \boldsymbol{\theta}_{T-1}\}$ and only introduces additional parameters $\boldsymbol{W}$, $\boldsymbol{b}$, and $\alpha$ as a router and weight shared between all layers. During fine-tuning, parameters in $\text{MoLEx}(f^{(0)})$ are updated to adapt to a downstream task via maximizing the conditional language modeling objective defined in Section 2.1 above.

To clarify our method's implementation, we insert the relevant parameter efficient fine-tuning method into the pre-trained model to obtain each layer $u_j$. Then, we initialize a trainable gate, $g$, in the model to be shared across all layers. This gate determines the top-1 layer selected, $v_t$, to be mixed with $u_j, j = 0, 1, \ldots, T-1$. We provide a diagram in Figure 1 for visualization of MoLEx.

The design of the proposed MoLEx transform in Eqn. 4 is based on the following three criteria:

**(1) Preserving the useful information in the pre-trained model:** In order to preserve the information in the pre-trained model, MoLEx reuses the trained layers in the pre-trained model to form a mixture of experts at each layer. Furthermore, at each layer $t$, we fix one expert to be the original layer $u_t$ and use the router $\boldsymbol{g}$ to select another expert, i.e., we use Top1 gating function.

Let us examine an example of a *2-layer linear backbone model* to illustrate how MoLEx preserves information in the pre-trained model. The backbone model $f^{(0)}$ in this case has the following form:

$$\boldsymbol{z}_0 = \boldsymbol{x}; \ \boldsymbol{z}_1 = \boldsymbol{z}_0 + \boldsymbol{W}_0\boldsymbol{z}_0; \ \boldsymbol{z}_2 = \boldsymbol{z}_1 + \boldsymbol{W}_1\boldsymbol{z}_1 = \boldsymbol{z}_0 + \boldsymbol{W}_0\boldsymbol{z}_0 + \boldsymbol{W}_1(\boldsymbol{z}_0 + \boldsymbol{W}_0\boldsymbol{z}_0).$$

(a) Naive model

(b) MoLEx model

Figure 1: **(a)** A naive parameter efficient fine-tuning model with $T$ layers, $u_0, u_1, \cdots, u_{T-1}$ and input $z_0$. $z_t$, for $t = 1, 2, \cdots, T$ are the outputs of each layer. **(b)** A MoLEx model transformed from a parameter efficient fine-tuning model with $T$ layers, $u_0, u_1, \cdots, u_{T-1}$ and input $z_0$. $z_t$, for $t = 1, 2, \cdots, T$ are the outputs of each MoLEx layer. At each layer, the input to the layer is processed by a gate $g$ to select the top-1 layer expert and the outputs of the layer and the selected layer are linearly combined and weighted by $\alpha$ and $1 - \alpha$ respectively. In the diagram, at layer $u_1$, layer $u_{T-1}$ is chosen by the gate for mixing. Then, the outputs of layer $u_1$ and layer $u_{T-1}$ are summed after multiplying them with $\alpha$ and $1 - \alpha$ respectively.

MoLEx($f^{(0)}$) can then be rewritten as

$$z_0 = x; z_1 = z_0 + \alpha W_0 z_0 + (1 - \alpha) v_0(z_0),$$

$$
\begin{aligned}
z_2 &= z_1 + \alpha W_1 z_1 + (1 - \alpha) v_1(z_1) \\
&= z_0 + \alpha W_0 z_0 + (1 - \alpha) v_0(z_0) + \alpha W_1(z_0 + \alpha W_0 z_0 + (1 - \alpha) v_0(z_0)) + (1 - \alpha) v_1(z_1) \\
&= \alpha \underbrace{(z_0 + W_0 z_0 + W_1(z_0 + W_0 z_0))}_{f^{(0)}} + (1 - \alpha) \underbrace{(z_0 + v_0(z_0) + v_1(z_1))}_{f^{(0)}_{\text{upcycled}}} + R,
\end{aligned}
$$

where the remainer $R = (1 - \alpha)\alpha W_1(v_0(z_0) - W_0 z_0)$, $z_0 = x$, and $v_t(z_t)$, $t = 1, 2$ is defined as in Eqn. 4. As can be seen in equation above, MoLEx($f^{(0)}$) is comprised of the pre-trained model $f^{(0)}$ and the additional upcycled part $f^{(0)}_{\text{upcycled}}$. The former component, $f^{(0)}$, allows MoLEx($f^{(0)}$) to maintain the information in the original pre-trained model.

**(2) Obtaining compositional representations:** At each layer, MoLEx combines the original layer $u_t$ and a layer $v_t$ as in Eqn. 4. Since $v_t$ is chosen among layers in the pre-trained model $f^{(0)}$ by the Top1 gating function, we can rewrite a layer $t$ of MoLEx($f^{(0)}$) as

$$z_{t+1} = z_t + \alpha u_t(z_t; \theta_t^{(0)}) + (1 - \alpha) u_\tau(z_t; \theta_\tau^{(0)}), \tag{5}$$

where $\tau \in \{0, 1, \ldots, T - 1\}$. To investigate the compositional representation captured by MoLEx, we distinguish between two cases: $\tau \geq t$ and $\tau < t$.

Case 1, $\tau \geq t$ (combining with a current/later layer): We apply Taylor expansion and approximate the processing at layer $\tau$ as follows:

$$u_\tau(z_\tau; \theta_\tau^{(0)}) = u_\tau\left(z_t + \sum_{j=t}^{\tau-1} u_j(z_j; \theta_j^{(0)}); \theta_\tau^{(0)}\right) \approx u_\tau(z_t; \theta_\tau^{(0)}) + u'_\tau(z_t; \theta_\tau^{(0)}) \sum_{j=t}^{\tau-1} u_j(z_j; \theta_j^{(0)}).$$

As can be seen, at layer $\tau$, $u_\tau$ implicitly extracts features from $z_t$ via the term $u_\tau(z_t; \theta_\tau^{(0)})$. Since $\tau > t$, these features are coarse-scale/high-level features of $z_t$ while the term $u_t(z_t; \theta_t^{(0)})$ in Eqn. 5 extracts the fine-scale/low-level features of $z_t$. Layer $t$ of MoLEx($f^{(0)}$) combines these coarse-scale/high-level and fine-scale/low-level features to attain a multi-scale compositional representation of the input data.

Case 2, $\tau < t$ (combining with a previous layer): In this case, we apply Taylor expansion to approximate $u_\tau(z_t; \theta_\tau^{(0)})$ as

$$u_\tau(z_t; \theta_\tau^{(0)}) = u_\tau\left(z_\tau + \sum_{j=\tau}^{t-1} u_j(z_j; \theta_j^{(0)}); \theta_\tau^{(0)}\right) \approx u_\tau(z_\tau; \theta_\tau^{(0)}) + u'_\tau(z_\tau; \theta_\tau^{(0)}) \sum_{j=\tau}^{t-1} u_j(z_j; \theta_j^{(0)}).$$

Here, $z_t = z_\tau + \sum_{j=\tau}^{t-1} u_j(z_j; \theta_j^{(0)})$, and $u_\tau$ extract finer-scale/lower-level features from $z_t$ via the terms $u_\tau, u_{\tau+1}, \ldots, u_{t-1}$ applied on $z_\tau, z_{\tau+1}, \ldots, z_{t-1}$, respectively. Layer $t$ of MoLEx($f^{(0)}$) combines these finer-scale/lower-level features with the coarse-scale/high-level features $u_t(z_t; \theta_t^{(0)})$ as in Eqn. 5 to achieve a multi-scale compositional representation of the input data.

**(3) Maintaining high efficiency:** Even though MoLEx introduces an additional layer expert at each layer, the increase in the total number of parameters of the sparsely upcycled model due to the router $g$ is negligible since MoLEx reuses layers from the pre-trained models (see Table 6). Moreover, at each layer, experts in MoLEx can be processed in parallel across different GPUs. Thus, the runtime of the MoLEx sparse upcycled model is comparable to the original model (see Table 6). Finally, from our experiments, we observe that setting $K > 1$ for the TopK gating function in Eqn. 4 does not yield a significant improvement in the model's performance (see Table 9). Thus, we set $K = 1$ in our design of MoLEx.

Despite its simple formulation, MoLEx offers an efficient and effective approach to sparse upcycling the models. Next, we will discuss the robustness property of MoLEx as an ensemble model.

## 2.3 MoLEx as an Ensemble Model

In this section, we consider the simple case when $u_j$ is a linear layer to provide insights into the advantages of MoLEx. We start with deriving an ensemble perspective of MoLEx from the linearity of each $u_j$ by unrolling $z_t$ to obtain

$$u_j(z_t) = u_j(z_{t-1} + \alpha u_{t-1}(z_{t-1}) + (1-\alpha)u_{i_{t-1}}(z_{t-1}))$$
$$= u_j(z_{t-1}) + \alpha u_j(u_{t-1}(z_{t-1})) + (1-\alpha)u_j(u_{i_{t-1}}(z_{t-1})).$$

We denote $i_t$ to be the layer index of the layer expert chosen by the gate at each layer $t$, i.e., to clarify, $u_{i_{t-1}} = v_{t-1}$ in Eqn. 4. Repeating this for each $j = 0, \cdots, T - 1$, in Eqn. 6, we write $z_{t+1}$ as a linear combination of compositions of $u_j$ weighted by $c_{i_0,i_1,\cdots,i_t} \geq 0$, a constant that is non-zero if and only if the combination $u_{i_t} \circ u_{i_{t-1}} \circ \cdots \circ u_{i_0}$ was chosen by the gate. We can re-label each sequence of $i_0, i_1, \cdots, i_t$ to an integer $j \in \{1, 2, \cdots, 3^{t+1} - 1\}$ and each composition of $u_{i_t} \circ u_{i_{t-1}} \circ \cdots \circ u_{i_0}$ to $f_j$ for $c_{i_0,i_1,\cdots,i_t} > 0$ as there are at most $3^{t+1} - 1$ combinations in $t$ layers of MoLEx.[1] Then, we will have

$$z_{t+1} = z_t + \alpha u_t(z_t) + (1-\alpha)u_{i_t}(z_t) \tag{6}$$

$$= z_0 + \sum_{T-1 \geq i_0,i_1,\cdots,i_t \geq 0} c_{i_0,i_1,\cdots,i_t} u_{i_t} \circ u_{i_{t-1}} \circ \cdots \circ u_{i_0}(x) = x + \sum_{j=1}^{3^{t+1}-1} c_j f_j$$

With such an unrolling, we are able to view a linear MoLEx model as an ensemble of linear models. Next, we will show that MoLEx, as an ensemble, is more robust than a single base model in the ensemble. We begin with a formal definition of robustness.

**Definition 1** ($\epsilon$-Robustness). *Consider an input $x$ and a classifier model, $f : \mathbb{R}^d \to [C]$, for a $C$-way classification task where $[C] = \{1, \cdots, C\}$. If for all $\tilde{x}$ within a closed ball of radius $\epsilon > 0$ with center $x$, i.e. $\tilde{x} \in B(x, \epsilon) = \{x + \delta : \|\delta\|_2 \leq \epsilon\}$, $f(\tilde{x}) = f(x)$, then we say $f$ is $\epsilon$-robust at $x$. We say that $f$ is more robust than $g$ if and only if $f$ is $\epsilon'$-robust and $g$ is $\epsilon$-robust at $x$, with $\epsilon' > \epsilon$.*

**Definition 2** (Linear MoLEx as an Ensemble Model). *From Eqn. 6, we can view a linear MoLEx model as a weighted ensemble of base functions, $f_j$, where each $f_j = u_{i_t} \circ u_{i_{t-1}} \circ \cdots \circ u_{i_0}$ is a composition of a certain permutation of the layers $u_t$, $t \in \{0, \cdots, T - 1\}$. For simplicity, let $f_0 = Id$, the identity function, $c_0 = 1$ and $n_t = 3^{t+1} - 1$, so that we can write $z_{t+1} = \sum_{j=0}^{n_t} c_j f_j$ as a MoLEx model with $t + 1$ layers.*

We consider a set of fine-tuning sample data $X$ drawn from some distribution $\chi$ with labels $Y$. For the ease of understanding, we consider the output of the MoLEx model, $z_{t+1}$, and a single base model with sequential layers, $f_{[0:t]} = u_t \circ u_{t-1} \circ \cdots u_0$, to be in the probability simplex, $\Delta^C = \{(x_1, x_2, \cdots, x_C) \in \mathbb{R}_{\geq 0}^C | \sum_{j=1}^C x_j = 1\}$ and refer to these as prediction models. A classifier model is then a prediction model composed with a classifier head $H(x) = \arg\max_i x_i$ where $x_i$ are

---

[1] As we unroll $z_t = z_{t-1} + (1-\alpha)u_t(z_{t-1}) + \alpha u_{i_t}(z_{t-1})$, we split each $u_j$ into 3 more $u_{j_1}, u_{j_2}, u_{j_3}$ terms at each layer and the skip connection $z_t$ into a $z_{t-1}$ and 2 more $u_{t_1}, u_{t_2}$ terms. Hence, at each $t$, we unroll 3 terms per term giving us $3^{t+1}$ from layer $t$ but 1 term will always unroll to the skip connection term $z_t$ until we have $z_0$. Hence, we subtract away this term to count the number of $u_j$ terms.

the elements of the vector $\boldsymbol{x}$. Then, our classifier model is $F(\boldsymbol{x}) = H(f(\boldsymbol{x})) = \arg\max_{i \in [C]} f(\boldsymbol{x})_i$ where $f(\boldsymbol{x})_i$ is the $i$-th element in the output vector $f(\boldsymbol{x})$.

It is not difficult to see that for an input vector $\boldsymbol{x} \in \boldsymbol{X}$ with label $y$, and a perturbed $\tilde{\boldsymbol{x}} \in B(\boldsymbol{x}, \epsilon)$, for a classifier $F = H(f)$ to remain $\epsilon$-robust at $\boldsymbol{x}$, we require that the prediction function satisfies

$$f(\tilde{\boldsymbol{x}})_y \geq f(\tilde{\boldsymbol{x}})_{y_i}, \forall y_i \neq y \tag{7}$$

where $f(\tilde{\boldsymbol{x}})_{y_i}$ is the $y_i$-th element of $f(\tilde{\boldsymbol{x}})$. Equivalently, we state this as a lemma below.

**Lemma 1** (Robustness condition for classifier model). *Consider a prediction function $f$, classifier head $H$, data point $(\boldsymbol{x}, y) \in (\boldsymbol{X}, \boldsymbol{Y})$ and a perturbed point $\tilde{\boldsymbol{x}} \in B(\boldsymbol{x}, \epsilon)$. If $F(\boldsymbol{x}) = H(f(\boldsymbol{x})) = y$, then $F$ is $\epsilon$-Robust at $\boldsymbol{x}$ if and only if*

$$\forall y_i \in [C], y_i \neq y, \min_{\tilde{\boldsymbol{x}} \in B(\boldsymbol{x}, \epsilon)} f(\tilde{\boldsymbol{x}})_y - f(\tilde{\boldsymbol{x}})_{y_i} \geq 0 \tag{8}$$

We are now ready to state our result regarding the improved robustness of linear ensembles and we defer all proofs to the appendix in section A.

**Theorem 1** (Linear ensembles are more robust). *Consider a data point $(\boldsymbol{x}, y) \in (\boldsymbol{X}, \boldsymbol{Y})$, $\epsilon > 0$, and $M$ linear base models, $f_j(\boldsymbol{x}) = \boldsymbol{W}_j^\top \boldsymbol{x}$ such that $\forall y_i$ and $\boldsymbol{W}_j$,*

*1. $\frac{1}{\epsilon}(\boldsymbol{e}_y - \boldsymbol{e}_{y_i})^\top f_j(\boldsymbol{x}) \geq \|\boldsymbol{W}_j(\boldsymbol{e}_y - \boldsymbol{e}_{y_i})\|_2$*

*2. $\boldsymbol{W}_j(\boldsymbol{e}_y - \boldsymbol{e}_{y_i})$ are not colinear,*

*where $e_y$ is the standard basis vector with $1$ at the $y$-th position and $0$ everywhere else. An ensemble classifier model, with a classification head $H$, $F_M = H(\sum_{j=0}^{M-1} c_j f_j)$ is $\epsilon'$-robust at $\boldsymbol{x}$ with $\epsilon' > \epsilon$.*

**Corollary 1** (Sufficient conditions for $\epsilon$-robustness). *Consider a data point $(\boldsymbol{x}, y) \in (\boldsymbol{X}, \boldsymbol{Y})$, if a classifier model $F = H(f)$ with prediction function, $f(\boldsymbol{x}) = \boldsymbol{W}^\top \boldsymbol{x}$ satisfies*

$$\frac{1}{\epsilon}(\boldsymbol{e}_y - \boldsymbol{e}_{y_i})^\top f(\boldsymbol{x}) \geq \|\boldsymbol{W}(\boldsymbol{e}_y - \boldsymbol{e}_{y_i})\|_2,$$

*then $F$ is $\epsilon$-robust at $\boldsymbol{x}$.*

**Corollary 2** (Linear MoLEx is more robust than sequential model). *If the base models of MoLEx $f_j = u_{i_t} \circ u_{i_{t-1}} \circ ... \circ u_{i_0}$ satisfies assumptions 1 and 2 in Theorem 1 above, then $\boldsymbol{z}_{t+1} = \sum_{j=0}^{n_t} c_j f_j$ is more robust than $f_{[0:t]}$.*

Consequently, we have established the robustness of a linear MoLEx model under perturbations within a closed $\epsilon$-ball.

## 3 EXPERIMENTAL RESULTS

In this section, we empirically validate the fine-tuning performance of MoLEx on the Natural Language Understanding (NLU) task, GLUE (Wang et al., 2018), the Natural Language Generation (NLG) benchmark, the End-to-End (E2E) dataset (Novikova et al., 2017a), and in a zero-shot evaluation on several GLUE tasks. Across all tasks and models, we apply MoLEx to LoRA on various models, including RoBERTa-base, RoBERTa-large (Liu, 2019), and GPT-2 (medium) (Radford et al., 2019). We use LoRA as our baseline for comparison. While MoLEx is compatible with any other fine-tuning method, we choose LoRA as it is one of the most popular light-weight adapters. Details on these tasks, models, metrics and implementations can be found in Appendix B. Our results are averaged over 5 runs with different seeds and conducted on a server with 8 A100 GPUs.

### 3.1 NATUAL LANGUAGE UNDERSTANDING

**GLUE** covers a wide range of domains, data types and challenge levels, making it a comprehensive benchmark for the generalizational ability of a language model. Using a pre-trained RoBERTa-base model from the HuggingFace Transformers library (Wolf et al., 2020), we fine-tune the models for all tasks using LoRA and MoLEx for comparison. To demonstrate the scalability of MoLEx to larger models, we also include RoBERTa-large and report our results in Table 1. Across all metrics, higher numbers indicate better performance.

In Table 1, we include results from prior works of other adaptation methods for reference. Details on each method can be found in the related work discussion in Section 5. As we implement MoLEx for

Table 1: RoBERTa-base (RoB$_{base}$) and RoBERTa-large (RoB$_{large}$) fine-tuned on the popular GLUE benchmark with different adaptations methods. MoLEx (bold and shaded in gray) is our proposed method in combination with LoRA. Hence, we use LoRA as our baseline and only reproduce results for LoRA in the table. An * indicates numbers published in previous work. For all tasks, we report accuracy except for Matthew's correlation for CoLA, Pearson correlation for STS-B, the overall (matched and mismatched) accuracy for MNLI. The average stated for models fine-tuned from the best MNLI checkpoint is the average of all tasks with results for MRPC, RTE and STS-B from the pre-trained RoBERTa checkpoint replaced by those from the MNLI checkpoint. Across almost all tasks, MoLEx surpasses the baseline LoRA on both both RoBERTa-base and RoBERTa-large, establishing its effectiveness and scalability.

| Model & Method | # Trainable Parameters | MNLI | SST-2 | MRPC | CoLA | QNLI | QQP | RTE | STS-B | Avg. |
|---|---|---|---|---|---|---|---|---|---|---|
| *Results published in prior works for reference* | | | | | | | | | | |
| RoB$_{base}$ (FT)* | 125.0M | 87.6 | 94.8 | 90.2 | 63.6 | 92.8 | 91.9 | 78.7 | 91.2 | 86.4 |
| RoB$_{base}$ (BitFit)* | 0.1M | 84.7 | 93.7 | 92.7 | 62.0 | 91.8 | 84.0 | 81.5 | 90.8 | 85.2 |
| RoB$_{base}$ (Adpt$^D$)* | 0.3M | 87.1$_{\pm.0}$ | 94.2$_{\pm.1}$ | 88.5$_{\pm1.1}$ | 60.8$_{\pm.4}$ | 93.1$_{\pm.1}$ | 90.2$_{\pm.0}$ | 71.5$_{\pm2.7}$ | 89.7$_{\pm.3}$ | 84.4 |
| RoB$_{base}$ (Adpt$^D$)* | 0.9M | 87.3$_{\pm.1}$ | 94.7$_{\pm.3}$ | 88.4$_{\pm.1}$ | 62.6$_{\pm.9}$ | 93.0$_{\pm.6}$ | 90.6$_{\pm.0}$ | 75.9$_{\pm2.2}$ | 90.3$_{\pm.1}$ | 85.4 |
| *Reproduced result from pre-trained RoBERTa checkpoint* | | | | | | | | | | |
| RoB$_{base}$ (LoRA) | 0.3M | 87.5$_{\pm.2}$ | 95.0$_{\pm.1}$ | 88.7$_{\pm.3}$ | 62.8$_{\pm1.0}$ | 93.2$_{\pm.2}$ | 90.8$_{\pm.0}$ | 76.9$_{\pm1.1}$ | 90.8$_{\pm.2}$ | 85.7 |
| **RoB$_{base}$ (MoLEx)** | **0.309M** | **87.7**$_{\pm.2}$ | **95.4**$_{\pm.2}$ | **89.8**$_{\pm.2}$ | **64.8**$_{\pm.5}$ | **93.2**$_{\pm.2}$ | **91.0**$_{\pm.0}$ | **77.3**$_{\pm1.3}$ | **91.0**$_{\pm.2}$ | **86.3** |
| *RoB$_{large}$ (LoRA)* | 0.8M | 90.7$_{\pm.1}$ | 96.3$_{\pm.2}$ | 90.9$_{\pm.4}$ | 67.8$_{\pm1.7}$ | 94.8$_{\pm.3}$ | 91.5$_{\pm.1}$ | 86.5$_{\pm.9}$ | 91.9$_{\pm.1}$ | 88.8 |
| **RoB$_{large}$ (MoLEx)** | 0.8M | **90.9**$_{\pm.1}$ | **96.4**$_{\pm.2}$ | **91.4**$_{\pm.7}$ | **68.2**$_{\pm.2}$ | 94.8$_{\pm.0}$ | **91.6**$_{\pm.1}$ | **87.1**$_{\pm.9}$ | **92.0**$_{\pm.2}$ | **89.1** |
| *Reproduced result from fine-tuned MNLI checkpoint* | | | | | | | | | | |
| RoB$_{base}$ (LoRA) | 0.3M | - | - | 89.7$_{\pm.6}$ | - | - | - | 86.8$_{\pm.2}$ | 91.3$_{\pm.1}$ | 87.1 |
| **RoB$_{base}$ (MoLEx)** | 0.3M | - | - | **91.1**$_{\pm.6}$ | - | - | - | 86.8$_{\pm.2}$ | 91.3$_{\pm.0}$ | 87.6 |

the LoRA adaptor, we focus on that method for comparison. We observe that across almost all tasks, MoLEx outperforms the baseline LoRA on both RoBERTa-base and RoBERTa-large, demonstrating the effectiveness and scalability of our method. A key advantage of MoLEx is its enhancement of model performance without any changes to the existing method or any increase in effective parameter count. Instead, it introduces a structural modification to the model's architecture, enabling the model to extract more information from the data, thereby leading to improved results.

## 3.2 Natural Language Generation

To further illustrate the versatility of our method on different language tasks, we evaluate MoLEx on the standard **E2E NLG Challenge** dataset introduced by (Novikova et al., 2017b) for training end-to-end, data-driven NLG systems. We fine-tune GPT-2 medium on E2E, following the set up of Li & Liang (2021), and report our results in Table 2. For all metrics, higher is better.

Similar to Table 1, in Table 2, we also include results from previous works. This is for reference, and we describe those methods in more detail in Section 5. Compared with the baseline LoRA method, MoLEx outperforms significantly on 3 metrics with a remarkable increase on BLEU by 0.7. We further note that the standard deviations for MoLEx is generally lower than LoRA. This aligns with our analysis of MoLEx as an ensemble model, which is expected to have lower variance (Ganaie et al., 2022; Gupta et al., 2022), and improves the reliability of the model in language generation.

## 3.3 Zero-shot transfer learning

We assess the ability of LoRA and MoLEx to transfer knowledge across relatively similar tasks in a zero-shot transfer learning setup on GLUE using RoBERTa-base. In Table 3, we present an evaluation of MoLEx in comparison with the baseline LoRA method when fine-tuned on one task and evaluated on another without any additional training. These pairs of tasks are QQP and MRPC (both test for semantic similarity), QQP and QNLI (both involve parsing questions), and QNLI and RTE (both are inference tasks). For all tasks, as they are binary classifications, when necessary, we reverse the class labels on the classifier head to obtain the best accuracy. In doing so, we are consistent across both models.

Table 3 suggests that MoLEx can generalize better to new data distributions compared to LoRA as across all evaluations, mixing layers consistently leads to significant improvements in zero-shot

Table 2: GPT2 medium (M) fine-tuned on the standard E2E NLG Challenge benchmark. We reproduce the LoRA baseline and compare it to our proposed method MoLEx (bold and shaded in gray) using the usual BLEU, NIST, MET, ROUGE-L and CIDEr metrics, where higher numbers indicate better performance. An * indicates numbers published in previous work and we include them in the table for reference. MoLEx significantly outperforms the baseline LoRA on 3 metrics with lower standard deviations, verifying its advantage.

| Model & Method | # Trainable Parameters | E2E NLG Challenge | | | | |
|---|---|---|---|---|---|---|
| | | BLEU | NIST | MET | ROUGE-L | CIDEr |
| *Results published in prior works for reference* | | | | | | |
| GPT-2 M (FT)* | 354.92M | 68.2 | 8.62 | 46.2 | 71.0 | 2.47 |
| GPT-2 M (Adapter$^L$)* | 0.37M | 66.3 | 8.41 | 45.0 | 69.8 | 2.40 |
| GPT-2 M (Adapter$^L$)* | 11.09M | 68.9 | 8.71 | 46.1 | 71.3 | 2.47 |
| GPT-2 M (Adapter$^H$)* | 11.09M | $67.3_{\pm.6}$ | $8.50_{\pm.07}$ | $46.0_{\pm.2}$ | $70.7_{\pm.2}$ | $2.44_{\pm.01}$ |
| GPT-2 M (FT$^{Top2}$)* | 25.19M | 68.1 | 8.59 | 46.0 | 70.8 | 2.41 |
| GPT-2 M (PreLayer)* | 0.35M | 69.7 | 8.81 | 46.1 | 71.4 | 2.49 |
| *Results reproduced for comparison* | | | | | | |
| GPT-2 M (LoRA) | 0.35M | $70.0_{\pm.5}$ | $8.77_{\pm.05}$ | $\mathbf{46.8}_{\pm.2}$ | $71.6_{\pm.3}$ | $2.52_{\pm.01}$ |
| **GPT-2 M (MoLEx)** | 0.359M | $\mathbf{70.7}_{\pm.4}$ | $\mathbf{8.87}_{\pm.03}$ | $46.5_{\pm.09}$ | $\mathbf{71.8}_{\pm.1}$ | $2.52_{\pm.01}$ |

Table 3: Zero-shot evaluation of RoBERTa-base on several GLUE tasks, QNLI, RTE, MRPC, and QQP when fine-tuned with LoRA and our MoLEx (bold and shaded in gray) on different tasks. As we only consider pairs of similar tasks for meaningful comparison, we report the relevant ones and mark the others with a dash. For all pairs of tasks considered, MoLEx outperforms the baseline LoRA by a large margin.

| Fine-tune on | Evaluate on | | | | | | | |
|---|---|---|---|---|---|---|---|---|
| | QNLI | | RTE | | MRPC | | QQP | |
| | LoRA | **MoLEx** | LoRA | **MoLEx** | LoRA | **MoLEx** | LoRA | **MoLEx** |
| QNLI | | | $56.7_{\pm1.1}$ | $\mathbf{59.9}_{\pm1.3}$ | - | - | $63.2_{\pm.0}$ | $\mathbf{65.7}_{\pm.0}$ |
| RTE | $56.1_{\pm.2}$ | $\mathbf{58.5}_{\pm.2}$ | | | - | - | - | - |
| MRPC | - | - | - | - | | | $65.7_{\pm.0}$ | $\mathbf{67.9}_{\pm.0}$ |
| QQP | $50.5_{\pm.2}$ | $\mathbf{56.2}_{\pm.2}$ | - | - | $67.2_{\pm.4}$ | $\mathbf{69.9}_{\pm.7}$ | | |

performance on new tasks. These results illustrate the ability of MoLEx to improve the model's transferability between different classification tasks, further validating our approach.

## 4 EMPIRICAL ANALYSIS

We conduct probing on MoLEx, additional experiments on robustness and efficiency, and an ablation study to provide more understandings of MoLEx.

### 4.1 PROBING TASKS

Language models, such as RoBERTa (Liu, 2019), attain impressive results on a multitude of NLP tasks that range in complexity, even with fine-tuning on a small subset of parameters (Zaken et al., 2021; Rücklé et al., 2021; Xu et al., 2023; Pfeiffer et al., 2021; Lin et al., 2020; Houlsby et al., 2019; Li & Liang, 2021). This suggests that the pre-trained base model already captures important linguistic properties of sentences that are capitalized upon during training on different tasks. At this junction, MoLEx with its unique feature of layer mixing can be leveraged to shed light on how the linguistic properties captured in the pre-trained base model can be combined for different downstream finetuning tasks.

We analyze the semantic nature captured by the representations in each layer of RoBERTa using the probing tasks proposed in (Conneau et al., 2018) and following the setup in (Jawahar et al., 2019). *For each probe, an auxiliary classification task is set up where the representations are used as features to predict certain linguistic properties of interest. The better the performance of the classifier, the more likely that the layer's hidden embedding encodes for that particular property.* These results are presented in Table 4. By piecing together the type of information mixed in each layer of MoLEx, we enhance our understanding of the language processing occurring in a RoBERTa model during fine-tuning and improve the interpretability of neural networks in NLP (Belinkov & Glass, 2019). We will focus on CoLA (single-sentence), STS-B (similarity and paraphrase) and RTE (inference) as representative tasks and examine the layers chosen for mixing to understand the key features that enable the model to excel on each type of task.

Table 4: Probing task performance (accuracy of a simple MLP classifier) for each layer of RoBERTa-base. Bolded numbers are the top 2 values within each task.

| Layer | SentLen (Surface) | WC (Surface) | TreeD (Syntactic) | TopConst (Syntactic) | BShift (Syntactic) | Tense (Semantic) | SubjNum (Semantic) | ObjNum (Semantic) | SOMO (Semantic) | CoordInv (Semantic) |
|---|---|---|---|---|---|---|---|---|---|---|
| 0 | **91.48** | **4.10** | **32.00** | 48.93 | 50.00 | 82.27 | 77.56 | 73.81 | 49.87 | 57.47 |
| 1 | **87.99** | 0.61 | 29.75 | 35.10 | 54.32 | 79.74 | 74.05 | 71.83 | 49.87 | 50.00 |
| 2 | 87.03 | 0.33 | 29.06 | 29.32 | 64.99 | 82.06 | 78.51 | 73.49 | 49.88 | 50.00 |
| 3 | 85.78 | 0.16 | 29.30 | 29.26 | 73.29 | 82.29 | 76.14 | 74.69 | 50.07 | 50.00 |
| 4 | 85.32 | 2.40 | 31.06 | 54.12 | 77.95 | 84.37 | 77.33 | 73.67 | 59.21 | 57.69 |
| 5 | 84.15 | 1.97 | **31.83** | 57.57 | 81.82 | 85.35 | 80.80 | 78.53 | 62.74 | 60.05 |
| 6 | 82.17 | 2.91 | 31.81 | **59.90** | 82.41 | 85.61 | 81.22 | **81.48** | 63.67 | 61.97 |
| 7 | 79.75 | 0.68 | 28.99 | 48.44 | 82.34 | 84.79 | 80.28 | 80.26 | **64.94** | 57.88 |
| 8 | 80.49 | 1.09 | 30.73 | 52.24 | **83.56** | 86.81 | 81.65 | 80.92 | **65.00** | 65.07 |
| 9 | 77.75 | 1.06 | 29.83 | 49.96 | 83.10 | 86.19 | 81.63 | 79.14 | 64.52 | **66.28** |
| 10 | 66.65 | 1.15 | 26.97 | 43.68 | 82.59 | 85.25 | 80.91 | 75.95 | 61.78 | 61.92 |
| 11 | 73.69 | **18.25** | 30.56 | **60.26** | **85.25** | **87.55** | **82.92** | 79.51 | 63.52 | **66.62** |

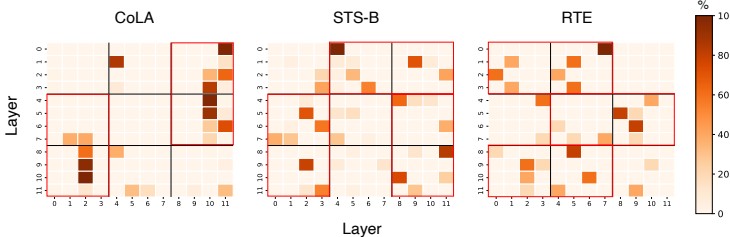

Figure 2: Heat maps to visualize the percentage of time each layer expert is chosen at every layer of MoLEx when fine-tuning RoBERTa-base on GLUE tasks, CoLA, STS-B and RTE. As one expert is fixed to be the original layer, the x-axis corresponds to the sequential layer while the y-axis corresponds to the layer experts.

The 10 probes can be grouped into 3 different categories, surface, syntactic, and semantic information tasks. Briefly, sentence length (SentLen) and word content (WC) fall under surface level information; the bigram shift (BShift), tree depth (TreeD) and top constituent (TopConst) tasks represent syntactic information; and the final 5 tasks, Tense, Subject Number (SubjNum), Object Number (ObjNum), semantic odd man out (SOMO) and coordination inversion (CoordInv) are considered semantic information. Detailed explanations on each task and their implementations can be found in the Appendix C. We provide our probing results for RoBERTa in Table 4 and discuss these results in detail in Appendix C.4.

**Key Linguistic Features for Task Performance.** In Figure 2, from left to right, there is an increasing degree of mixing between all layers that correlates with the increasing complexity of each task. In particular, since RTE is an inference task that requires a deeper understanding of the input sentences, it is not surprising that MoLEx mixes nearly all layers to a greater extent for this task than for both CoLA and STS-B. We discuss the probing for each task below.

**CoLA** evaluates grammatical acceptability and, as shown in Figure 2, focuses on later layers, which capture semantic information like tense and word placement—key for grammatical correctness. **STS-B** measures sentence similarity, with Figure 2 showing significant mixing in later layers for rich contextual data and earlier layers for surface features like sentence length, reflecting task-specific needs. **RTE**, a binary entailment classification task, emphasizes middle layers, aligning with the syntactic structure required for logical understanding. These observations suggest the model adapts its layer usage to the linguistic demands of each task.

## 4.2 ROBUSTNESS

Though the models used in our experiments are nonlinear, we expect that the theoretical robustness properties still hold and can be extended to practical situations. To verify this, we perform a simple experiment using MoLEx and LoRA in a RoBERTa-base model trained on 2 GLUE tasks as described in Section 3 and present the results in Table 5. For the tasks presented, we add random noise into the input data for evaluation and find that MoLEx is indeed more robust to noise than the baseline

Table 5: Robustness (in accuracy) of RoBERTa-base on GLUE tasks, QNLI, and SST2 when fine-tuned with LoRA and MoLEx. Random noise is added to the input during evaluation. MoLEx is more robust than the LoRA baseline.

| Method | QNLI | SST-2 |
|---|---|---|
| | (with added noise) | |
| LoRA | 63.1 $_{+.2}$ | 69.3 $_{+.1}$ |
| MoLEx | **64.0** $_{+.2}$ | **70.9** $_{+.2}$ |

Table 6: Run time per sample, memory, and number of parameters for baseline LoRA and our MoLEx in RoBERTa-base during inference time.

| Method | Sec/Sample (Inference) | Memory (Inference) | Parameters |
|---|---|---|---|
| *LoRA (baseline)* | 0.00345 | 1890MB | 0.3M |
| MoLEx | 0.00357 | 1892MB | 0.3M |

LoRA model as it achieves a higher accuracy on both tasks. We do not report the results for the other tasks, as adding noise causes both models to have an accuracy equivalent to random guessing.

### 4.3 EFFICIENCY ANALYSIS & ABLATION STUDY

We provide the run time per sample, memory, and number of parameters of MoLEx compared to the baseline LoRA in Table 6 and a more detailed analysis in Appendix E.3. There is only a marginal increase in compute time due to the additional gating function. Also, in Table 9, Appendix D, we conduct 3 GLUE tasks, CoLA, QQP, and SST-2 when using Top1 and Top2 routing. We observe that Top1 yields better results. Thus, we use a Top1 routing for MoLEx.

## 5 RELATED WORK

**Parameter-Efficient Fine-Tuning (PEFT).** The simplest solution of PEFT is to only update a small subset of weights (partial fine-tuning) (Li & Liang, 2021). For comparison, we include results from a previous work that kept all layers except the last 2 frozen on GPT-2 in Table 2 (FT^Top2) (Li & Liang, 2021). Other methods that fine-tune a selected subset of parameters include BiTFiT (Zaken et al., 2021), where only the bias vectors are updated, and its extension using Neural Architecture Search (Lawton et al., 2023). A separate approach is to introduce extra trainable parameters into the model for adaptation. These include soft prompt-based tuning where trainable word embeddings are inserted among the input tokens (Hambardzumyan et al., 2021; Lester et al., 2021; Liu et al., 2023; Zhang et al., 2023) or prepended to the hidden states of the multi-head attention layer (prefix-tuning) (Li & Liang, 2021). Another method is prefix-layer tuning (PreLayer) that learns new activations after every Transformer layer. Qi et al. (2022) suggests only training the gain and bias term of the LayerNorm in the model. In addition, adapter tuning (Houlsby et al., 2019) involves inserting adapter layers into a transformer layer. This design is denoted as Adapter^H in Table 2. More efficient methods have also been proposed by (Lin et al., 2020; Pfeiffer et al., 2021) to reduce the number of adapter layers (Adapter^L) and by (Rücklé et al., 2021) to drop adapter layers (Adapter^D). Recently, neural functional networks have emerged as a promising alternative to PEFT (Zhou et al., 2023a;b; Mitchell et al., 2022; Vo et al., 2024; Sinitsin et al., 2020; Navon et al., 2023; Tran et al., 2025a;b).

**Neural Network Intepretability.** The study of how models learn language structure during training is gaining interest. (Belinkov & Glass, 2019). Specifically, there is interest in deciphering the type of linguistic knowledge encoded in sentence and word embeddings (Dalvi et al., 2017; Belinkov et al., 2017; 2018; Sennrich, 2017). Many studies focus on uncovering the structural properties of language captured by BERT (Devlin et al., 2018) mainly through various linguistic probes on the representations produced by the model (Devlin et al., 2018; Liu et al., 2019; Tenney et al., 2019; Hewitt & Manning, 2019; Conneau et al., 2018) and well-designed evaluation protocols and stimuli (Goldberg, 2019; Marvin, 2018; Gulordava, 2018; Linzen et al., 2016). There is also a general consensus that language models learn linguistic information hierarchically (Peters et al., 2018).

## 6 CONCLUDING REMARKS

In this paper, we introduce a Mixture of Layer Experts (MoLEx), a novel approach that leverages layers as experts to facilitate the exchange of linguistic information and improve a model's fine-tuning and transfer knowledge ability. Orthogonal to current PEFT methods, we do not add in or modify any internal components in the model. Instead, we propose a structural change to the architecture of the model that can be effortlessly integrated with any PEFT method while maintaining the same number of effective parameters. We theoretically justify the robustness of MoLEx in a simplified model and provide empirical evidence for it. Our experiments demonstrate that MoLEx significantly improves performance across a range of downstream tasks, including the GLUE benchmark and the E2E Challenge, while incurring minimal additional computational overhead and scales well with model size. Additionally, MoLEx's unique architecture also enhances model interpretability. A limitation of our work is that our robustness guarantee is only for deep linear models. Extending this result to the case of deep nonlinear models, as well as exploring layer mixing across different models, is an interesting direction to pursue. We leave these exciting research ideas as future work.

ACKNOWLEDGMENTS

This research / project is supported by the National Research Foundation Singapore under the AI Singapore Programme (AISG Award No: AISG2-TC-2023-012-SGIL). This research / project is supported by the Ministry of Education, Singapore, under the Academic Research Fund Tier 1 (FY2023) (A-8002040-00-00, A-8002039-00-00). This research / project is also supported by the NUS Presidential Young Professorship Award (A-0009807-01-00) and the NUS Artificial Intelligence Institute–Seed Funding (A-8003062-00-00).

**Reproducibility Statement.** Source code for our experiments are provided in the supplementary material. We provide the full details of our experimental setup – including datasets, model specification, train regime, and evaluation protocol – for all experiments Section 3 and Appendix B. All datasets are publicly available.

**Ethics Statement.** Given the nature of the work, we do not foresee any negative societal and ethical impacts of our work.

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

# Supplement to "MoLEx: Mixture of Layer Experts for Finetuning with Sparse Upcycling"

**Table of Contents**

## A  PROOFS

### A.1  PROOF OF THEOREM 1

We restate the theorem below for convenience.

**Theorem 1** (Linear ensembles are more robust than base models)**.** *For a data point* $(\boldsymbol{x}, y) \in (\boldsymbol{X}, \boldsymbol{Y})$, *and* $M$ *linear base models,* $f_j(\boldsymbol{x}) = \boldsymbol{W}_j^\top \boldsymbol{x}$ *such that* $\forall y_i$ *and* $\boldsymbol{W}_j$,

    *1.* $\frac{1}{\epsilon}(\boldsymbol{e}_y - \boldsymbol{e}_{y_i})^\top f_j(\boldsymbol{x}) \geq \|\boldsymbol{W}_j(\boldsymbol{e}_y - \boldsymbol{e}_{y_i})\|_2$

    *2.* $\boldsymbol{W}_j(\boldsymbol{e}_y - \boldsymbol{e}_{y_i})$ *are not colinear,*

*an ensemble classifier model, with a classification head* $H$, $F_M = H(\sum_{j=0}^{M-1} c_j f_j)$ *is* $\epsilon'$-*robust at* $\boldsymbol{x}$ *with* $\epsilon' > \epsilon$.

*Proof.* For a linear ensemble classifier, $F_M$ to be robust, from Lemma 1, we require that $\forall y_i \in [C], y_i \neq y, \min_{\tilde{\boldsymbol{x}} \in B(\boldsymbol{x}, \epsilon)} \sum_{j=0}^{M-1} c_j(f_j(\tilde{\boldsymbol{x}})_y - f_j(\tilde{\boldsymbol{x}})_{y_i}) \geq 0$. Expanding this, with $\boldsymbol{e}_y$ being the

standard basis vector with 1 in the $y$-th position,

$$\min_{\tilde{\boldsymbol{x}} \in B(\boldsymbol{x}, \epsilon)} \sum_{j=0}^{M-1} c_j (f_j(\tilde{\boldsymbol{x}})_y - f_j(\tilde{\boldsymbol{x}})_{y_i})$$

$$= \min_{\tilde{\boldsymbol{x}} \in B(\boldsymbol{x}, \epsilon)} \sum_{j=0}^{M-1} c_j (\boldsymbol{e}_y - \boldsymbol{e}_{y_i})^\top f_j(\tilde{\boldsymbol{x}})$$

$$= \min_{\tilde{\boldsymbol{x}} \in B(\boldsymbol{x}, \epsilon)} \sum_{j=0}^{M-1} c_j (\boldsymbol{e}_y - \boldsymbol{e}_{y_i})^\top (\boldsymbol{W}_j^\top \boldsymbol{x} + \boldsymbol{W}_j^\top (\tilde{\boldsymbol{x}} - \boldsymbol{x}))$$

$$= \sum_{j=0}^{M-1} c_j (\boldsymbol{e}_y - \boldsymbol{e}_{y_i})^\top f_j(\boldsymbol{x}) + \min_{\tilde{\boldsymbol{x}} \in B(\boldsymbol{x}, \epsilon)} (\boldsymbol{e}_y - \boldsymbol{e}_{y_i})^\top \left( \sum_{j=0}^{M-1} c_j \boldsymbol{W}_j^\top \right) (\tilde{\boldsymbol{x}} - \boldsymbol{x})$$

$$= \sum_{j=0}^{M-1} c_j (\boldsymbol{e}_y - \boldsymbol{e}_{y_i})^\top f_j(\boldsymbol{x}) + \min_{\tilde{\boldsymbol{x}} \in B(\boldsymbol{x}, \epsilon)} (\bar{\boldsymbol{W}}(\boldsymbol{e}_y - \boldsymbol{e}_{y_i}))^\top (\tilde{\boldsymbol{x}} - \boldsymbol{x})$$

$$\geq \sum_{j=0}^{M-1} c_j (\boldsymbol{e}_y - \boldsymbol{e}_{y_i})^\top f_j(\boldsymbol{x}) - \epsilon \| \bar{\boldsymbol{W}}(\boldsymbol{e}_y - \boldsymbol{e}_{y_i}) \|_2$$

where the last inequality holds by the Cauchy-Schwartz inequality and we denote $\bar{\boldsymbol{W}}^\top := (\sum_{j=0}^{M-1} c_j f_j) = \sum_{j=0}^{M-1} c_j \boldsymbol{W}_j^\top$ to represent our ensemble function. Hence, if the following holds,

$$\sum_{j=0}^{M-1} c_j (\boldsymbol{e}_y - \boldsymbol{e}_{y_i})^\top f_j(\boldsymbol{x}) - \epsilon \| \bar{\boldsymbol{W}}(\boldsymbol{e}_y - \boldsymbol{e}_{y_i}) \|_2 \geq 0 \iff \frac{1}{\epsilon} \sum_{j=0}^{M-1} c_j (\boldsymbol{e}_y - \boldsymbol{e}_{y_i})^\top f_j(\boldsymbol{x}) \geq \| \bar{\boldsymbol{W}}(\boldsymbol{e}_y - \boldsymbol{e}_{y_i}) \|_2,$$

then $F_M$ is robust. Since, in our assumption 2, $\forall y_i$ and $\boldsymbol{W}_j$, $\boldsymbol{W}_j(\boldsymbol{e}_y - \boldsymbol{e}_{y_i})$ are not colinear, from triangle inequality and assumption 1, we have

$$\| \bar{\boldsymbol{W}}(\boldsymbol{e}_y - \boldsymbol{e}_{y_i}) \|_2 = \| \sum_{j=0}^{M-1} c_j \boldsymbol{W}_j(\boldsymbol{e}_y - \boldsymbol{e}_{y_i}) \|_2 < \sum_{j=0}^{M-1} c_j \| \boldsymbol{W}_j(\boldsymbol{e}_y - \boldsymbol{e}_{y_i}) \|_2$$

$$\leq \frac{1}{\epsilon} \sum_{j=0}^{M-1} c_j (\boldsymbol{e}_y - \boldsymbol{e}_{y_i})^\top f_j(\boldsymbol{x})$$

As the inequality holds strictly, we can always find an $\epsilon' > \epsilon$ such that the inequality still holds. Hence, $F_M$ is $\epsilon'$-robust. $\square$

## A.2 Proof of Corollary 1

We restate the corollary below for convenience.

**Corollary 1** (Sufficient conditions for $\epsilon$-robustness). *For a data point $(\boldsymbol{x}, y) \in (\boldsymbol{X}, \boldsymbol{Y})$, if a classifier model $F = H(f)$ with prediction function, $f(\boldsymbol{x}) = \boldsymbol{W}^\top \boldsymbol{x}$ satisfies $\frac{1}{\epsilon}(\boldsymbol{e}_y - \boldsymbol{e}_{y_i})^\top f(\boldsymbol{x}) \geq \| \boldsymbol{W}(\boldsymbol{e}_y - \boldsymbol{e}_{y_i}) \|_2$, then $F$ is $\epsilon$-robust at $\boldsymbol{x}$.*

*Proof.* This result follows directly from the proof of Theorem 1, with $M = 1$. $\square$

## A.3 Proof of Corollary 2

We restate the corollary below for convenience.

**Corollary 2** (Linear MoLEx is more robust than sequential model). *If the base models of MoLEx $f_j = u_{i_t} \circ u_{i_{t-1}} \circ \dots \circ u_{i_0}$ satisfies assumptions 1 and 2 in Theorem 1 above, then $\boldsymbol{z}_{t+1} = \sum_{j=0}^{n_t} c_j f_j$ is more robust than $f_{[0:t]}$.*

*Proof.* In each layer $t$ of MoLEx, as one layer expert is always fixed to be the original pre-trained layer $u_t$, the sequential model, $f_{[0:t]}$ will always be one of the base models. Then, by Corollary 1 and assumption 1, $f_{[0:t]}$ is $\epsilon$-robust. The rest of the corollary follows as a consequence of Theorem 1 as $\boldsymbol{z}_{t+1}$ will be $\epsilon'$-robust with $\epsilon' > \epsilon$. $\square$

# B  ADDITIONAL EXPERIMENTAL DETAILS

## B.1  NATURAL LANGUAGE UNDERSTANDING: GLUE

**Tasks:** CoLA (Warstadt et al., 2019) consists of sequences of words taken from books and journal articles on linguistic theory with labels to determine if they are grammatically acceptable or not. SST-2 (Socher et al., 2013) comprises of movie reviews and the task is to predict their sentiments as positive or negative. MRPC (Dolan & Brockett, 2005) is a corpus of pairs of sentences pulled from online news sources and annotated by humans whether they are semantically equivalent. The QQP[2] dataset was collated from the community question-answering website Quora. It contains question pairs and simlar to MRPC, the goal is to determine if they are labelled to be semantically equivalent. STS-B (Cer et al., 2017) is another sentence pair similarity task extracted from news headlines, video and image captions, and natural language inference data. However, it differs from the previous tasks in not using binary labels and instead each examples is accompanied by a similarity score from 1 to 5. MNLI (Williams et al., 2018) uses pairs of premise and hypothesis sentences that have been collected from ten different sources, including transcribed speech, fiction, and government reports. The objective is to predict whether the premise entails the hypothesis (entailment), contradicts the hypothesis (contradiction), or neither (neutral). QNLI (Rajpurkar et al., 2018) is a task to determine if the context sentence in a question-sentence pair contains the answer to the question. The sentences were taken from paragraphs in Wikipedia and the questions were annotated by humans. Lastly, we have RTE (Dagan et al., 2006; Bar-Haim et al., 2006; Giampiccolo et al., 2007; Bentivogli et al., 2009), a compilation of datasets from a series of annual textual entailment challenges. Similar to MNLI, the objective is to determine if the sentence pairs contain an entailment or not. The classes for contradiction and neutral as in MNLI are collapsed into a single non-entailment class.

**Metrics:** All tasks in GLUE are classification tasks, except for STS-B which is a regression task. Therefore, the metric reported for STS-B is the Pearson correlation coefficient as is standard practise. We report the overall accuracy for MNLI which includes both matched and mismatched data. These correspond to evaluations on pairs of sentences within the same domain or cross-domain respectively. On CoLA, we use the Matthews correlation coefficient (Matthews, 1975) for evaluation due to the unbalanced binary classification data. This metric ranges from -1 to 1, with 0 indicating random guessing. For all other tasks, we present their accuracy for evaluation. Across all metrics, a higher number reflects stronger performance.

**Model:** We use the pre-trained RoBERTa-base and RoBERTa-large model (Liu, 2019) from the HuggingFace Transformers library (Wolf et al., 2020) for evaluation on the GLUE task. RoBERTa is an optimized version of the original pre-training recipe proposed in BERT (Devlin et al., 2018). RoBERTa-base has 125M parameters with 12 layers, 12 attention heads and 768 hidden dimensions while RoBERTa-large has 355M parameters with 24 layers, 16 attention heads and 1024 hidden dimensions.

**Implementation details:** We follow the same fine-tuning set up as in the original LoRA (Hu et al., 2021) paper for all GLUE experiments using the their publicly available code `https://github.com/microsoft/LoRA`. We use the same setting for fine-tuning on the pre-trained model and from an MNLI checkpoint. For each task, we also optimize the hyperparameters of the gate used in deciding the layer experts to be used for mixing. These settings can be found in Table 7 and for all gates, we use the same optimizer, AdamW (Loshchilov, 2017), as the LoRA parameters with a learning rate of 0.1 and weight decay of 0.01. We report the mean and standard deviation over 5 random seeds for all results and the result for each run is taken from the best epoch.

While we employ batch routing in the mixture of layers, each token will have a different choice of layer to be routed to as every token is processed by the gate. In deciding the overall batch's decision, we use 2 different aggregates. The first is a majority-takes-all scheme where we route the batch to the layer which majority of tokens have chosen. The second is to use the maximum over the mean probability vector of all the tokens choices. These are referred to as Mode and Mean respectively under Batch Agg in the table. For gate types with suffix "Sig" we use a sigmoid activation before taking TopK values and the default is a softmax activation. For almost all gates, if they do not have an "Indv Gate", this means that we use the same gate for all layers to decide the mixing layers. On RTE and STS-B, we use individual gates, which means that each layer has its own linear gating function and mixing weights instead of sharing one between all the layers. For all tasks, if the mixing weights are fixed, we use $\alpha = 0.95$ as defined in Eqn. 4.

Table 7: Hyperparameter settings for LoRA and MoLEx on each GLUE task when fine-tuning RoBERTa-base and RoBERTa-large.

| Method | Dataset | MNLI | SST-2 | MRPC | CoLA | QNLI | QQP | RTE | STS-B |
|---|---|---|---|---|---|---|---|---|---|
| | Optimizer | | | | AdamW | | | | |
| | Warmup Ratio | | | | 0.06 | | | | |
| | LR Schedule | | | | Linear | | | | |
| RoBERTa-base LoRA | Batch Size | 16 | 16 | 16 | 32 | 32 | 16 | 32 | 16 |
| | # Epochs | 30 | 60 | 30 | 80 | 25 | 25 | 80 | 40 |
| | Learning Rate | 5E-04 | 5E-04 | 4E-04 | 4E-04 | 4E-04 | 5E-04 | 5E-04 | 4E-04 |
| | LoRA Config. | | | | $r_q = r_v = 8$ | | | | |
| | LoRA $\alpha$ | | | | 8 | | | | |
| | Max Seq. Len. | | | | 512 | | | | |
| RoBERTa-base MoLEx gate | Gate Type | Cos-Sig | Cos | Linear | Linear | Cos | Cos-Sig | Linear | Linear |
| | Projection Dim | 416 | 128 | - | - | 96 | 384 | - | - |
| | Indv Gate | × | × | × | × | × | × | ✓ | ✓ |
| | Batch Agg | Mode | Mode | Mode | Mode | Mean | Mode | Mean | Mean |
| | Mixing Weights | Learn | Learn | Learn | Fix | Learn | Learn | Fix | Fix |
| | Load Balance | 0.005 | 0.01 | 0.0 | 0.01 | 0.001 | 0.001 | 0.001 | 0.006 |
| RoBERTa-large LoRA | Batch Size | 4 | 4 | 4 | 4 | 4 | 4 | 8 | 8 |
| | # Epochs | 10 | 10 | 20 | 20 | 10 | 20 | 20 | 30 |
| | Learning Rate | 3E-04 | 4E-04 | 3E-04 | 2E-04 | 2E-04 | 3E-04 | 4E-04 | 2E-04 |
| | LoRA Config. | | | | $r_q = r_v = 8$ | | | | |
| | LoRA $\alpha$ | | | | 16 | | | | |
| | Max Seq. Len. | 128 | 128 | 512 | 128 | 512 | 512 | 512 | 512 |
| RoBERTa-large MoLEx gate | Gate Type | Cos | Cos | Linear | Linear | Cos | Cos-Sig | Linear | Linear |
| | Projection Dim | 416 | 64 | - | - | 256 | 384 | - | - |
| | Indv Gate | × | × | ✓ | × | × | × | ✓ | ✓ |
| | Batch Agg | Mode | Mode | Mode | Mode | Mean | Mode | Mode | Mode |
| | Mixing Weights | Fix | Fix | Fix | Fix | Fix | Learn | Fix | Fix |
| | Load Balance | 0.0001 | 0.0 | 0.0001 | 0.01 | 0.001 | 0.001 | 0.0 | 0.0 |

## B.2 NATURAL LANGUAGE GENERATION: E2E

**Dataset:** The E2E NLG dataset approximately consists of more than 50,000 examples from the restaurant domain and there is a 76.5-8.5-15 split of the dataset into a training, validation and test set respectively. The E2E dataset is commonly used for the evaluation of data-to-text tasks and brings new challenges such as open vocabulary, complex syntactic structures and diverse discourse phenomena. Every data input consists of a meaning representation (MR) that includes a sequence of attribute-value pairs and a corresponding target, a natural language (NL) reference text.

**Metrics:** We report the same metrics as in (Novikova et al., 2017b), namely BLEU (Papineni et al., 2002), NIST (Doddington, 2002), METEOR (Lavie & Agarwal, 2007), ROUGE-L (Lin, 2004) and CIDEr (Vedantam et al., 2015). BLEU is a method to evaluate the quality of automated machine translations that scales the geometric mean of the precision scores of the n-grams in a generated text by an exponential brevity penalty factor. Similarly, NIST is based on BLEU with some slight changes. NIST uses weighted precision scores of the n-grams determined by how informative each of them are, instead of an equal weighting as in BLEU, and loosens the brevity penalty for small variations. METEOR evaluates the quality of the generated text at a segment level. It constructs a word alignment between strings and scores them using a parameterized harmonic mean of their unigram precision and recall. ROGUE-L is a metric that naturally captures sentence level structures by only awarding scores to in-sequence co-occurrences in the predicted and reference text. Lastly, CIDEr is a measure for how well the generated text matches the consensus of a set of reference image descriptors. It scores the frequency of n-grams in the generated text that occurs in the reference sentences and discounts n-grams that appear commonly across all images in the dataset.

**Model:** We use the pre-trained GPT-2 medium (Radford et al., 2019) from the HuggingFace Transformers library (Wolf et al., 2020) for evaluation on the E2E dataset. GPT-2 medium contains 355M parameters with 24 layers, 16 attention heads and 1,024 hidden dimensions.

Table 8: Hyperparameter settings for LoRA and MoLEx on the E2E NLG task when fine-tuning GPT-2 medium (M).

|  | Dataset | E2E |
|---|---|---|
|  | *Training* |  |
| GPT-2 M LoRA | Optimizer | AdamW |
|  | Weight Decay | 0.01 |
|  | Dropout Prob | 0.1 |
|  | Batch Size | 8 |
|  | # Epoch | 5 |
|  | Warmup Steps | 500 |
|  | Learning Rate Schedule | Linear |
|  | Label Smooth | 0.1 |
|  | Learning Rate | 0.0002 |
|  | Adaptation | $r_q = r_v = 4$ |
|  | LoRA $\alpha$ | 32 |
| GPT-2 M MoLEx gate | Gate Type | Linear |
|  | Layers with MoLEx | 0 to 11 (inclusive) |
|  | Indv Gate | $\times$ |
|  | Batch Agg | Mode |
|  | Mixing Weights | Fixed |
|  | Load Balance | 0.01 |
|  | *Inference* |  |
|  | Beam Size | 10 |
|  | Length Penalty | 0.9 |
|  | No Repeat Ngram Size | 4 |

**Implementation details:** We follow the same fine-tuning setup as in Li & Liang (2021) and LoRA (Hu et al., 2021) using their publicly available code `https://github.com/microsoft/LoRA`. We also optimize the hyperparameters of the gate used in deciding the layer experts to be used for mixing. These settings can be found in Table 8 and we use the same optimizer, AdamW (Loshchilov, 2017), as the LoRA parameters with a learning rate of 0.1 and weight decay of 0.01. We report the mean and standard deviation over 5 random seeds for all results and the result for each run is taken from the best epoch.

While we employ batch routing in MoLEx, each token will have a different choice of layer to be routed to as every token is processed by the gate. In deciding the overall batch's decision for GPT-2, we use a majority-takes-all scheme where we route the batch to the layer which majority of tokens have chosen (Mode). We use a linear gating function with a softmax activation and only implement MoLEx in the first 12 layers of the model. The mixing weights are fixed and we use a value of $\alpha = 0.95$ as defined in Eqn. 4. All layers share the same gate for routing.

## C    ADDITIONAL EMPIRICAL ANALYSIS DETAILS

### C.1    PROBING TASKS

Probing (or diagonostic) tasks (Adi et al., 2016; Hupkes et al., 2018; Conneau et al., 2018) aid us in the discovery of linguistic features potentially encoded in a deep learning model. Specifically, in the hidden representations of the input in each layer. In order to understand these representations using a probe, an auxiliary classification task is set up where the representations are used as features to predict certain linguistic properties of interest. The better the performance of the classifier, the more likely that the layer's hidden embedding encodes for that particular property. Using the 10 probing tasks developed by (Conneau et al., 2018) and inspired by (Jawahar et al., 2019), who had done a similar analysis on BERT, we evaluate each layer of RoBERTa and present the results in Table 4.

In each tasks's dataset, there are 100K training sentences and 10K-sentence validation and test sets. All sets are equally balance among the target classes. These datasets were constructed by (Conneau et al., 2018) from the Toronto Book Corpus (Zhu, 2015; Paperno et al., 2016).

**Surface Information:** SentLen is a task to predict the length of a sentence, which is considered to be the number of words in the sentence. It is converted into a 6-way classification task by grouping sentence lengths into 6 equal-width bins. WC is a classification task with 1000 classes. Each class is a word and each input is a sentence that contains one and only one of the words within those classes. The task is to predict which word is contained within the input sentence.

**Syntactic Information:** BShift is a binary classification task where half the dataset has sentences intact and another half has sentences with 2 random adjacent words inverted. The goal is to predict if the sentence has a legal word order or if it has been inverted. TreeD assesses whether the hierarchical structure of sentences can be inferred from the hidden layer's embedding. The task is to determine the depth of the longest path from root to any leaf in the sentence, with possible depths ranging from 5 to 12. Hence, resulting in a 8-way classification task. TopConst is a 20-class task where 19 classes represent the most frequent top constituent sequence and the last class is for all the others. The classifier has to identify which sequence of top constituents immediately follow the input sentence node, which is illustrative of the latent syntactic structures captured by each layer's representation.

**Semantic Information:** The goal of the Tense task is to identify the tense of the main-clause verb in the input sentence. For the SubjNum and ObjNum tasks, both focus on the number of the subject and respectively, direct object, of the main clause. In the SOMO dataset, sentences are modified through the replacement of a random noun or verb with another in a challenging way. The bigrams containing these noun or verb replacements will have a comparable corpus frequency with the original, making the task all the more difficult. The last task, the CoordInv dataset comprises of sentences with pairs of coordinate clauses, of which some orders have been inverted. The classifier is meant to identify if the sentences are intact or inverted as a binary classification task.

## C.2    IMPLEMENTATION DETAILS

We use the SentEval toolkit (Conneau & Kiela, 2018), available publicly at `https://github.com/facebookresearch/SentEval`, and the same set up as (Jawahar et al., 2019) for our probe analysis. We send each of the datasets in the 10 probing tasks through the pre-trained RoBERTA-base model that we use for fine-tuning and extract the feature representations from each layer. Next, we train classifiers, that are simple MLPs with a sigmoid activation, on these features as input. We use the recommended hyperparameter search space of $\{50, 100, 200\}$ hidden units and $\{0.0, 0.1, 0.2\}$ dropout for each task and an additional logistic regression model for the word content (WC) task as it contains 1,000 classes. We report the best classifier's results in Table 4.

## C.3    FULL RESULTS OF SECTION 4

We present the full results of the different layer experts being mixed at inference time for all GLUE tasks when fine-tuning RoBERTa-base with MoLEx in Figure 3. Interestingly, for QQP and MNLI, there is a heavy emphasis on the middle layers. As the middle layers encode for syntactic information, MNLI as an inference task does require that structural information to understand the logical implications of the input. However, as QQP is a semantic similarity classification task, it is not obvious why it would require more syntactic information. Indeed, if we look at MRPC, a similar task on sentences instead of questions, it mainly chooses the earlier layers for surface level information which does make sense for sentence similarity. The main distinction between the 2 tasks is that the inputs are either questions or sentences. A plausible explanation is that questions require more syntactic information to be understood, resulting in our findings.

## C.4    LINGUISTIC PROPERTIES CAPTURED BY ROBERTA

In this section we will discuss the linguistic properties captured by RoBERTa as revealed through our probe analysis. We observe in Table 4 that across almost all probes, layer 11 does particularly well, suggesting that the last layer of the model encodes a considerable amount of general linguistic information. This is the main contrast to the probe analysis performed on BERT in (Jawahar et al., 2019) and could be an unintended consequence of the optimized training recipe in RoBERTa, highlighting how various training protocols can influence the learning outcomes of a model. It is also worth noting that all probes on RoBERTa, except for WC, performs roughly on the same scale as BERT while WC is much poorer in comparison, even with logistic regression. This could suggest that this surface level information is not relevant to the NLP in the model.

The remainder of the analysis corroborates with the probe analysis on BERT whereby the early layers contain superficial information, the middle layers, syntactic information and later layers, seman-

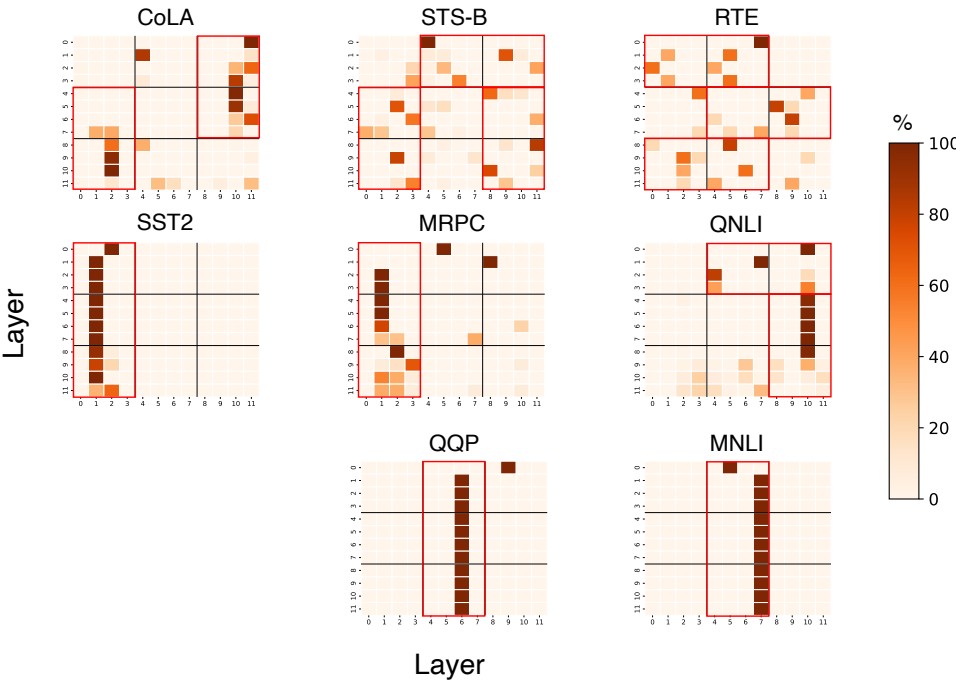

Figure 3: Plots of heat maps to visualize the percentage of time each layer expert is chosen at every layer of MoLEx when fine-tuning RoBERTa-base on all GLUE tasks. As one expert is fixed to be the original layer, the x-axis corresponds to the sequential layer while the y-axis corresponds to the layer experts. The darker a square is, the more often that layer is chosen by the gate during inference. For example, when fine-tuning on CoLA, layer 9 mixes with layer 2, 100% of the time. The grids are partitioned into thirds along the x-axis and y-axis for easy visualization of early, middle and later layers.

Table 9: Comparison of RoBERTa-base on GLUE tasks, CoLA, QQP and SST-2 when fine-tuned with MoLEx using Top-1 and Top-2 routing. We report accuracy for all tasks in the table below.

| Method | CoLA | QQP | SST-2 |
|---|---|---|---|
| MoLEx (Top1) | **64.8** $_{\pm.5}$ | **91.0** $_{\pm.0}$ | **95.4** $_{\pm.2}$ |
| MoLEx (Top2) | 63.7 $_{\pm.4}$ | 90.7 $_{\pm.0}$ | 95.0 $_{\pm.3}$ |

tic. This further aligns with the intuition that more complex structures within the data are revealed deeper in the model as it undergoes more processing as discussed in Section 2.2.

## D  ABLATION STUDY

Table 9 compares results on 3 GLUE tasks, CoLA, QQP, and SST-2 when using Top1 and Top2 routing. We observe that Top1 yields more improvement. Thus, we use a Top1 routing for our MoLEx models.

## E  ADDITIONAL EXPERIMENTAL RESULTS AND EFFICENCY ANALYSIS

### E.1  FULL PARAMETER FINE-TUNING FOR ROBERTA

We conduct additional experiments for RoBERTa-base using full parameter fine-tuning on the GLUE benchmark. We present the results in Table 10 below. Our MoLEx model consistently outperforms the full parameter fine-tuning across all tasks. These findings further confirm MoLEx's adaptability to different models and training methods.

### E.2  FINE-TUNING LLAMA-3.2-1B USING LORA

We conduct additional experiments to fine-tune Llama-3.2-1B on the Alpaca dataset using LoRA. We use the publicly available repository https://github.com/meta-llama/llama3 for our experiments and employ MoLEx to fine-tune the model in comparison with LoRA. As shown

Table 10: RoBERTa-base with full parameter fine-tuning and with MoLEx when fine-tuned on the GLUE benchmark. We report accuracy for all tasks except for, Pearson correlation for STS-B, Matthew's correlation for CoLA and the overall (matched and mismatched) accuracy for MNLI. A higher value reflects a better performance of the model.

| Method | RTE | MRPC | STS-B | CoLA | MNLI | QNLI | SST-2 | QQP | Ave. |
|---|---|---|---|---|---|---|---|---|---|
| RoBERTa (full parameter) | 80.1 | 88.7 | 90.9 | 62.6 | 87.7 | 92.9 | 95.0 | 91.8 | 86.2 |
| RoBERTa (MoLEx) | **80.9** | **89.5** | **91.1** | **63.3** | **87.8** | **93.1** | **95.1** | 91.8 | **86.6** |

Table 11: Train and validation perplexity (PPL) when fine-tuning Llama-3.2-1B on Alpaca using LoRA and LoRA + MoLEx. Lower PPL is indicative of better performance.

| Method | Train PPL ($\downarrow$) | Validation PPL ($\downarrow$) |
|---|---|---|
| LoRA | 4.18 | 4.11 |
| MoLEx | **4.05** | **4.02** |

Table 12: Accuracy when evaluating Llama-3.2-1B on MMLU, AGIEval English, Hellaswag, and ARC-Challenge using LoRA and LoRA + MoLEx. A higher value is indicative of better performance.

| Method | MMLU | AGIEval English | Hellaswag | ARC-Challenge |
|---|---|---|---|---|
| LoRA | 30.42 | 19.16 | 47.14 | 36.69 |
| MoLEx | **31.51** | **19.81** | **48.23** | **37.80** |

Table 13: Efficiency analysis of Llama-3.2-1B fine-tuned using LoRA during inference on the Alpaca dataset with and without MoLEx implemented.

| Method | Total Parameters | Trainable Parameters | Trainable Parameters (%) | Memory (MB) | Flop/ Sample | Sec/ Sample | Flop/ Sec | Min/ Epoch |
|---|---|---|---|---|---|---|---|---|
| LoRA | 1,236,666,368 | 851,968 | 0.0689 | 10,442 | 12.329 T | 0.506 | 24.366 T | 4:59 |
| MoLEx | 1,236,699,152 | 884,752 | 0.0715 | 10,442 | 22.511 T | 0.557 | 40.415 T | 6:00 |

in Table 11, on this task with Llama-3.2-1B, MoLEx achieves better train and validation PPL than LoRA, demonstrating the effectiveness of MoLEx in large language models.

Further, we evaluate each model on the standard MMLU (Hendrycks et al., 2020), AGIEval English (Zhong et al., 2024), Hellaswag (Zellers et al., 2019), and ARC-Challenge dataset Clark et al. (2018) and report their results in Table 12. Consistent with our results on Alpaca, MoLEx improves over the naive LoRA model, confirming its advantage.

### E.3    DETAILED EFFICIENCY ANALYSIS ON LLAMA-3.2-1B

While more resources are required during inference in MoLEx as compared to the naive PEFT model, these can be accelerated during inference time through parallization. In this section, we include a more detailed efficiency analysis when implementing MoLEx in Llama-3.2-1B and using LoRA for fine-tuning on the Alpaca dataset.

As computational efficiency refers to the amount of time required for a given step in a calculation, we maintain that MoLEx is as efficient as the original method used without MoLEx. While MoLEx almost doubles the overall computational load (flops), there is only a minimal increase in inference time due to parallelization of the forward computations through two layers. We present our analysis in the Table 13 for a more detailed comparison with naive PEFT models.

