# OpenReview forum: "MoLEx: Mixture of Layer Experts for Fine-tuning with Sparse Upcycling"
_ICLR.cc/2025/Conference — ICLR 2025 Poster_

### Official Review · Reviewer_UiHH · 2024-10-25

**Soundness:** 2
**Presentation:** 1
**Contribution:** 2
**Rating:** 5
**Confidence:** 4

**Summary:**

This paper presents MoLEx, a simple and effective method that does not alter the model's structure or add extra parameters. Instead, it treats all layers as experts and introduces a router at each layer to select the top k layers for activation. Extensive experiments on RoBERTa and GPT-2 demonstrate the strong performance of MoLEx.

**Strengths:**

This paper introduces a novel sMoE model whose experts are layers in the pre-trained model. MoLEx is a simple yet effective approach without additional expert parameters.

**Weaknesses:**

Weaknesses & Questions

1. The introduction of this paper does not effectively convey the motivation for the proposed MoLEx method. I would appreciate it if the authors could briefly clarify the limitations of existing sMoE methods and explain the specific problems that the proposed approach aims to address.

2. The methods section of this paper relies heavily on mathematical formulas, whereas the MoLEx method itself is relatively straightforward. I suggest that the authors consider providing a method diagram to complement the mathematical expressions, as this could facilitate a better understanding of the proposed method.

3. I believe there are some unclear aspects in the implementation of the method presented in this paper. Specifically, when mixing layers, is the paper using  $x_{t+1}=x_t+\alpha u_{t}(x_{t})+(1-\alpha) u_{l}(x_{t})$, where \$u_{l}$ is the top-1 layer selected by the router, to obtain $u_{t+1}$? Or is it treating the low-rank matrices $A$ and $B$ from LoRA as experts to derive $x_{t+1}$ through $x_{t+1}=x_{t}+u_{t}(x_{t})+\alpha B_{t}(A_{t}(x_{t})) + (1-\alpha) B_{l}(A_{l}(x_{t}))$? I also hope the authors can add relevant training details in the methods or implementation section to clarify this.

4. Referring to point three, I believe the authors' assessment of efficiency may be inaccurate regardless of the method used. The claim that "these layer experts can be computed in parallel and do not increase the computational time" (lines 494-495) seems misleading. Since each input must undergo forward computations through two layers in the case of top-1, the total computational load will inevitably increase, not just due to the additional computations from the gate layer. Therefore, it may not be appropriate to assert that there is no impact on efficiency solely based on parallel computation. I encourage the authors to reassess their evaluation of efficiency, particularly by considering results from single-GPU inference.

5. The descriptions of the trainable parameters for LoRA and MoLEx in Tables 1 and 2 may be misleading. It could lead readers to believe that MoLEx has no additional trainable parameters compared to LoRA. However, the inclusion of the gate layer, regardless of its parameter count, should be represented to accurately reflect the differences. I encourage the authors to provide a clear comparison of the trainable parameters in the tables.

6. The authors utilized eight A100 GPUs for their experiments, but models like RoBERTa and GPT-2 seem somewhat outdated at this point. I suggest that the authors consider expanding their experiments to include more recent LLMs, such as Qwen-2.5 or Llama-3.1, to enhance the credibility of their findings. Additionally, evaluating these models using popular current assessment methods like MMLU, GSM8K, and HumanEval would provide valuable insights.

7. The MoLEx method proposed in this paper appears to be adaptable to various models and training methods. However, it was only validated under the LoRA method. I suggest that the authors include experiments with full parameter training to provide a more comprehensive evaluation of the method's effectiveness.

8. Current sMoE models applied to LLMs typically operate at the token level, where different experts are activated for each input token. However, the appendix of this paper states that MoLEx employs a sentence-level approach, with all tokens in a batch going through the same experts. It would be valuable for the authors to further explore how MoLEx performs in a token-level activation context, as this could provide insights into its adaptability and effectiveness in such scenarios.

9. Line 148 of the paper concludes that "preserving the useful information in the pre-trained model" is achievable, supported by extensive mathematical derivations. However, I believe this conclusion is not adequately reflected in the experiments. As noted in [1], parameters themselves can retain knowledge, so a catastrophic forgetting experiment is necessary to validate this claim. I suggest using benchmarks like TRACE [2] to assess whether MoLEx can mitigate the issue of catastrophic forgetting.

[1] HFT: Half Fine-Tuning for Large Language Models

[2] TRACE: A Comprehensive Benchmark for Continual Learning in Large Language Models

**Questions:**

See Weaknesses.

---

> ### Author Response · Authors · 2024-11-19
> **Review Reply (1/4): Q1, Q2, Q3**
>
> **Q1. The introduction of this paper does not effectively convey the motivation for the proposed MoLEx method. I would appreciate it if the authors could briefly clarify the limitations of existing sMoE methods and explain the specific problems that the proposed approach aims to address.**
>
> **Answer:** Thanks for your comments. Please allow us to clarify the motivation of MoLEx.
>
> 1. On one hand, **MoLEx is a novel sparse mixture of experts (SMoE) whose experts are layers in a pre-trained model**. In particular, given a dense pre-trained model, MoLEx combines layers in the pre-trained model for fine-tuning on downstream tasks. Compared with other SMoE methods, **MoLEx does not introduce new expert parameters and is memory efficient**. In order to turn a dense pre-trained model into an SMoE, other existing SMoE methods introduces new experts with their own parameters at each layer, thus increasing the total number of parameters of the model. MoLEx reuses the layers in the dense pretrained models as additional experts at each layer. During fine-tuning, the parameters of these experts are shared with those of the corresponding layers in the dense model. Compared with the dense model, MoLEx only introduces new parameters for the routers, but not the experts. As the only additional parameters introduced is a single linear layer for the routing function, the memory required to load MoLEx onto a GPU or CPU is almost the same as the dense checkpoint and less than an SMoE model.
>
> 2. On another hand, **MoLEx is an innovative parameter efficient fine-tuning method**. By combining pre-trained layers, MoLEx enhances the model's predictive capabilities on downstream tasks, achieving better fine-tuning performance without increasing the effective number of parameters, as shown in Table 1 and 2 for accuracy enhancement, Table 3 for zero-shot evaluation, and Table 5 for robustness improvement, in our manuscript.
>
>
> **Q2. The methods section of this paper relies heavily on mathematical formulas, whereas the MoLEx method itself is relatively straightforward. I suggest that the authors consider providing a method diagram to complement the mathematical expressions, as this could facilitate a better understanding of the proposed method.**
>
> **Answer:** Thanks for your suggestion. While the concept of MoLEx is novel, the method itself is simple to understand and easy to implement, which is indeed an advantage of MoLEx. We rigorously formulate MoLEx in Section 2.2 of the main text to provide a theoretical guarantee of its improved robustness compared to the baseline model, as demonstrated in Theorem 1 in Section 2.3. Additionally, our mathematical formulation of MoLEx allows us to analyze MoLEx's capability to retain valuable information from the pre-trained model and achieve compositional representations later in Section 2.2. As the reviewer suggests, we have included a method diagram in Section 2.2, Figure 1 to demonstrate our proposed MoLEx method.
>
>
> **Q3. I believe there are some unclear aspects in the implementation of the method presented in this paper. Specifically, when mixing layers, is the paper using:
> $x_{t+1} = x_t + \alpha u_t(x_t) + (1 - \alpha) u_l(x_t)$, where $u_l$ is the top-1 layer selected by the router, to obtain $u_{t+1}$? Or is it treating the low-rank matrices $A$ and $B$ from LoRA as experts to derive $x_{t+1}$ through $x_{t+1} = x_t + u_t(x_t) + \alpha B_t(A_t(x_t)) + (1 - \alpha) B_l(A_l(x_t))$?
> I also hope the authors can add relevant training details in the methods or implementation section to clarify this.**
>
> **Answer:** We are using $x_{t+1} = x_t + \alpha u_t(x_t) + (1 - \alpha) u_l(x_t)$, where $u_l$ is the top-1 layer selected by the router. We only employ LoRA to adapt our MoLEx model to downstream tasks. We have added relevant training details in Section 2.2 of our manuscript to clarify this and summarize the training details below.
>
> To clarify the implementation of our method, we first insert the relevant parameter efficient fine-tuning method into the pre-trained model to obtain each layer $u_j$. Next, we initialize a trainable gate in the model, to be shared across all layers. This gate will determine the top-1 layer selected, $u_l$, to be mixed with $u_j$ at each $j=0,1,\cdots,T-1$.

---

> ### Author Response · Authors · 2024-11-19
> **Review Reply (2/4): Q4, Q5, Q6**
>
> **Q4 .Referring to point three, I believe the authors' assessment of efficiency may be inaccurate regardless of the method used. The claim that "these layer experts can be computed in parallel and do not increase the computational time" (lines 494-495) seems misleading. Since each input must undergo forward computations through two layers in the case of top-1, the total computational load will inevitably increase, not just due to the additional computations from the gate layer. Therefore, it may not be appropriate to assert that there is no impact on efficiency solely based on parallel computation. I encourage the authors to reassess their evaluation of efficiency, particularly by considering results from single-GPU inference.**
>
> **Answer:** We agree with the reviewer that each input must undergo forward computations through two layers, hence increasing the total computational load. However, as computational efficiency refers to the amount of time required for a given step in a calculation, we maintain that MoLEx is as efficient as the original method used without MoLEx. This is due to the parallelization of the two forward computations which maintains the computational time while the computational load is doubled. We provide a more detailed efficiency analysis in the Table 1 below and included it in Appendix E.3 of our revised manuscript.
>
> Table 1: Efficiency analysis of Llama-3.2-1B fine-tuned using LoRA during inference on the Alpaca dataset with and without MoLEx implemented.
>
>
> | Model \ Metric | Total Parameters | Trainable Parameters| Trainable Parameters (%) |Memory (MB)| Flop/Sample  | Sec/Sample | Flop/Sec |
> | -------- | -------- | -------- |-------- |-------- |-------- |-------- |-------- |
> | Llama-3.2-1B (Baseline) | 1,236,666,368  | 851,968  |0.0689  |10,442     | 12.329 T | 0.506 | 24.366 T |
> | Llama-3.2-1B (MoLEx)  | 1,236,699,152  | 884,752     |0.0715     |10,442     |22.511 T | 0.557 | 40.415 T |
>
> As we see in Table 1, while the number of flops per sample is almost doubled, the amount of time per sample is only slightly increased. Hence, MoLEx is as efficient as the naive parameter efficient fine-tuning model.
>
> Regarding single-GPU inference, it is also possible to compute two forward computations in parallel on a single-GPU due to the Pytorch asynchronous nature. We refer to a [reference in pytorch](https://d2l.ai/chapter_computational-performance/async-computation.html).
>
>
> **Q5. The descriptions of the trainable parameters for LoRA and MoLEx in Tables 1 and 2 may be misleading. It could lead readers to believe that MoLEx has no additional trainable parameters compared to LoRA. However, the inclusion of the gate layer, regardless of its parameter count, should be represented to accurately reflect the differences. I encourage the authors to provide a clear comparison of the trainable parameters in the tables.**
>
> **Answer:** We only use 1 gate network, a.k.a, router, for all layers in MoLEx, i.e., layers in MoLEx share the same gate network. The number of parameters of this gate network is 9,228, which is neglegible when compared with the total number of the trainable parameters in LoRA, which is 0.3M. We have updated Table 1 and 2 in our revision to include these additional parameter counts.
>
> **Q6. The authors utilized eight A100 GPUs for their experiments, but models like RoBERTa and GPT-2 seem somewhat outdated at this point. I suggest that the authors consider expanding their experiments to include more recent LLMs, such as Qwen-2.5 or Llama-3.1, to enhance the credibility of their findings. Additionally, evaluating these models using popular current assessment methods like MMLU, GSM8K, and HumanEval would provide valuable insights.**
>
> **Answer:** Thank you for your suggestion. As you suggested, we have conducted additional experiments to fine-tune Llama-3.2-1B on the Alpaca dataset using LoRA. We use the publicly available repository https://github.com/meta-llama/llama3 for our experiments and employ MoLEx to fine-tune the model in comparison with LoRA. As shown in Table 2 below, on this task with Llama-3.2-1B, MoLEx achieves better train and validation PPL than LoRA. These results further justify the effectiveness of our method. We have also included these results in Appendix E.2, Table 11, in our revision.
>
> Table 2: Train and validation perplexity (PPL) when fine-tuning Llama-3.2-1B on Alpaca using LoRA and LoRA + MoLEx. Lower PPL is indicative of better performance.
>
> | Model \ Metric| Train PPL ($\downarrow$) | Validation PPL ($\downarrow$) |
> | -------- | -------- | -------- |
> | LoRA  | 4.18    | 4.11     |
> | MoLEx  | **4.05**    | **4.02**   |

---

> > ### Comment · Reviewer_UiHH · 2024-11-21
> >
> > Thank you for the author's response. Regarding Q6, I believe that PPL is generally not used to evaluate a model's performance during the SFT stage. Instead, benchmarks like MMLU are typically used for evaluation. Could you provide additional relevant evaluation metrics?

---

> > > ### Author Response · Authors · 2024-11-21
> > > **Follow up on Q6**
> > >
> > > Thank you for your prompt reply. Following your suggestion, we have also run the evaluation on MMLU using Llama-3.2-1B. We report our results in Table 1 below. We improved by about 1% on MMLU, further verifying the effectiveness of MoLEx. We also include these results in Table 12 in Appendix E.2 of our revision.
> > >
> > > Table 1: Accuracy when evaluating Llama-3.2-1B on MMLU using LoRA and LoRA + MoLEx. A higher value is indicative of better performance.
> > > | Model \ Metric| Acc. % ($\uparrow$) |
> > > | -------- | -------- |
> > > | LoRA  | 30.42 |
> > > | MoLEx  | **31.51**    |

---

> > > > ### Author Response · Authors · 2024-11-24
> > > > **Follow up on Q6 (Part 2)**
> > > >
> > > > Following reviewer UiHH's suggestion, we evaluated Llama-3.2-1B on MMLU to compare LoRA and MoLEx. Following up on his suggestion, we ran 3 more evaluations, AGIEval English, Hellaswag, and ARC-Challenge, and present our results in Table 1. We include our previous result on MMLU in Table 1 for completeness.
> > > >
> > > > As observed from Table 1, we improved on all evaluations conducted, further verifying the effectiveness of MoLEx. We also include these results in Table 12 in Appendix E.2 of our revision.
> > > >
> > > > Table 1: Accuracy when evaluating Llama-3.2-1B on MMLU, AGIEval English, Hellaswag, and ARC-Challenge, using LoRA and LoRA + MoLEx. A higher value is indicative of better performance.
> > > > | Model \ Metric| MMLU | AGIEval English | Hellaswag | ARC-Challenge |
> > > > | -------- | -------- |-------- |-------- |-------- |
> > > > | LoRA  | 30.42 | 19.16|47.14 |36.69|
> > > > | MoLEx  | **31.51**    |**19.81**|**48.23**|**37.80**|

---

> > > > > ### Comment · Reviewer_UiHH · 2024-11-24
> > > > >
> > > > > Thank you for your additional experiments, I will raise the score.

---

> > > > > > ### Author Response · Authors · 2024-11-24
> > > > > > **Thanks for your consideration**
> > > > > >
> > > > > > Thank you for your response and support.
> > > > > >
> > > > > > We would greatly appreciate it if you could share any remaining concerns about our work so that we can address them before the rebuttal period concludes. We are more than happy to engage in follow-up discussions to resolve your concerns and kindly ask you to consider whether raising your score to 6 might better reflect your updated evaluation of our paper.
> > > > > >
> > > > > > Thank you once again for your time and thoughtful feedback!

---

> > ### Comment · Reviewer_UiHH · 2024-11-21
> >
> > Regarding Q4, I noticed that there is an approximate 10% increase in time during the inference phase. I’m curious about how much additional time overhead is needed during the training phase. Could you provide a rough time comparison?

---

> > > ### Author Response · Authors · 2024-11-21
> > > **Follow up on Q4**
> > >
> > > Thank you for your suggestion. Fine-tuning on the Alpaca dataset with Llama-3.2-1B using LoRA takes 4:59 minutes per epoch, whereas using MoLEx requires 6:00 minutes per epoch. We summarize these results in Table 2 below. Same results can be found in Table 13 in Appendix E.3 of our revised manuscript.
> > >
> > > Table 2: Time in minutes when training Llama-3.2-1B on the Alpaca dataset using LoRA and LoRA + MoLEx for 1 epoch.
> > > | Model \ Metric| min/epoch |
> > > | -------- | -------- |
> > > | LoRA  | 4:59 |
> > > | MoLEx  | 6:00   |

---

> ### Author Response · Authors · 2024-11-19
> **Review Reply (3/4): Q7, Q8**
>
> **Q7.The MoLEx method proposed in this paper appears to be adaptable to various models and training methods. However, it was only validated under the LoRA method. I suggest that the authors include experiments with full parameter training to provide a more comprehensive evaluation of the method's effectiveness.**
>
> **Answer:** Thank you for your comments. As the reviewer noted, MoLEx is adaptable to various models and training methods. Following the reviewer's suggestion, we have conducted additional experiments for RoBERTa using full parameter fine-tuning on 4 GLUE tasks. We present the results in Table 3 below. Our MoLEx consistently outperforms the full parameter fine-tuning across all tasks. These findings further confirm MoLEx's adaptability to different models and training methods. We have included these results in Appendix E.1, Table 10, in our revision. We are currently running additional results for this experiment on other GLUE tasks and will report the results in our next reply.
>
> Table 3: RoBERTa with full parameter fine-tuning and with MoLEx when fine-tuned on 4 GLUE tasks, RTE, MRPC, STS-B and CoLA. We report accuracy for RTE and MRPC, Pearson correlation for STS-B and Matthew's correlation for CoLA. A higher value reflects a better performance of the model.
>
> | Model \ Task| RTE | MRPC | STS-B |CoLA |
> | -------- | -------- | -------- |--|--|
> | RoBERTa (full parameter)  | 80.1    | 88.7     | 90.9 |62.6|
> | RoBERTa (MoLEx)  | **80.9**    | **89.5**   | **91.1**|**63.3**|
>
>
>
> **Q8. Current sMoE models applied to LLMs typically operate at the token level, where different experts are activated for each input token. However, the appendix of this paper states that MoLEx employs a sentence-level approach, with all tokens in a batch going through the same experts. It would be valuable for the authors to further explore how MoLEx performs in a token-level activation context, as this could provide insights into its adaptability and effectiveness in such scenarios.**
>
> **Answer:** We agree with the reviewer that investigating MoLEx's performance in a token-level setting would offer valuable insights into its adaptability and effectiveness. However, the implementation of this token-level approach can be inefficient due to the complexities involved in scattering and gathering the tokens to different experts. A highly optimized code written at the CUDA level is needed to efficiently implement the token-level approach for MoLEx. We leave this as future work.

---

> > ### Author Response · Authors · 2024-11-22
> > **Follow up on Q7**
> >
> > As the reviewer UiHH noted, MoLEx is adaptable to various models and training methods. Following the reviewer's suggestion, we conducted  experiments for RoBERTa using full parameter fine-tuning on 4 GLUE tasks, RTE, MRPC, STS-B and CoLA.
> >
> > We have finished running additional results for this experiment on the remaining GLUE tasks and report our full results below in Table 1. We include results on the previous tasks reported as well for convenience. The same results can be found in Table 10 in Appendix E.3 of our revised manuscript.
> >
> > Our MoLEx consistently surpasses full parameter fine-tuning on each task, highlighting its versatility across various models and training methods.
> >
> > Table 1: RoBERTa with full parameter fine-tuning and with MoLEx when fine-tuned on the GLUE benchmark. We report accuracy for all tasks except for, Pearson correlation for STS-B, Matthew's correlation for CoLA and the overall (matched and mismatched) accuracy for MNLI. A higher value reflects a better performance of the model.
> >
> > | Model \ Task| RTE | MRPC | STS-B |CoLA |MNLI | QNLI | SST-2 |QQP |Ave.|
> > | -------- | -------- | -------- |--|--|--|--|--|--|--|
> > | RoBERTa (full parameter)  | 80.1    | 88.7     | 90.9 |62.6| 87.7|92.9 | 95.0|91.8 |86.2|
> > | RoBERTa (MoLEx)  | **80.9**    | **89.5**   | **91.1**|**63.3**|**87.8** | **93.1**|**95.1** | 91.8| **86.6**|

---

> ### Author Response · Authors · 2024-11-19
> **Review Reply (4/4): Q9**
>
> **Q9. Line 148 of the paper concludes that "preserving the useful information in the pre-trained model" is achievable, supported by extensive mathematical derivations. However, I believe this conclusion is not adequately reflected in the experiments. As noted in [1], parameters themselves can retain knowledge, so a catastrophic forgetting experiment is necessary to validate this claim. I suggest using benchmarks like TRACE [2] to assess whether MoLEx can mitigate the issue of catastrophic forgetting.**
>
> **Answer:** We believe there is a misunderstanding of the advantage of our MoLEx method in preserving the useful information in the pre-trained model. Please allow us to clear this misunderstanding by clarifying that our MoLEx is designed for parameter-efficient finetuning, but not continual learning. By claiming that MoLEx "preserves the useful information in the pre-trained model", we mean that MoLEx, as a new parameter-efficient finetuning method, can preserve the useful information in the pre-trained model with its richer structure for obtaining better performance on a downstream task but not to mitigate the catastrophic forgetting. However, following the reviewer's suggestion, we have conducted an additional experiment on the TRACE benchmark and include our results in Table 4, 5 and 6 below.
>
> Table 4: Overall Performance (OP) and Backward Transfer Score (BWT) of Llama-2-7B on the continual learning TRACE benchmark when fine-tuned sequentially using LoRA and MoLEx.
> | Model \ Metric | OP ($\uparrow$) | BWT ($\uparrow$) |
> | -- | -- | -- |
> | LoRA   | 41.51    | -15.11  |
> | MoLEx   |39.43   | -14.98     |
>
> Table 5: Detailed results of the performance of Llama-2-7B at each round of the continual learning TRACE benchmark when fine-tuned sequentially using LoRA. We report accuracy for all tasks except ROUGE-L for MeetingBank, SARI for 20Minuten and Edim similarity for Py150. Each column represents the task that the model is trained on and the rows are the tasks that the model is evaluated on after being trained on the task in the that round.
>
> |                | C&#x2011;STANCE | FO&#x2011;C  | MeetingBank | Py150  | ScienceQA | NumGLUE&#x2011;cm | NumGLUE&#x2011;ds | 20Minuten |
> |----|---|---|---|--|---|--|--|--|
> | **C-STANCE**   | 49.25    | 44.3  | 42.7        | 32.35  | 0         | 41.15      | 37.5       | 36.4      |
> | **FOMC**       |          | 70.97 | 58.47       | 62.9   | 0         | 50.2       | 47.58      | 55.44     |
> | **MeetingBank**|          |       | 49.17       | 25.93  | 16.57     | 18.37      | 19.42      | 0         |
> | **Py150**      |          |       |             | 57.16  | 21.74     | 44.72      | 43.92      | 26.1      |
> | **ScienceQA**  |          |       |             |        | 72.65     | 62.6       | 59.15      | 55        |
> | **NumGLUE-cm** |          |       |             |        |           | 41.98      | 29.63      | 29.63     |
> | **NumGLUE-ds** |          |       |             |        |           |            | 55.38      | 44.62     |
> | **20Minuten**  |          |       |             |        |        |       |         | 40.83     |
>
> Table 6: Detailed results of the performance of Llama-2-7B at each round of the continual learning TRACE benchmark when fine-tuned sequentially using MoLEx. We report accuracy for all tasks except ROUGE-L for MeetingBank, SARI for 20Minuten and Edim similarity for Py150. Each column represents the task that the model is trained on and the rows are the tasks that the model is evaluated on after being trained on the task in the that round.
>
> |                | C&#x2011;STANCE | FOMC  | MeetingBank | Py150  | ScienceQA | NumGLUE&#x2011;cm | NumGLUE&#x2011;ds | 20Minuten |
> |----|---|---|---|--|---|--|--|--|
> | **C-STANCE**   | 48.1     | 44.5  | 43.6        | 39.9   | 12.2      | 39.4       | 39         | 38.27     |
> | **FOMC**       |          | 68.35 | 59.88       | 54.84  | 0         | 34.27      | 36.09      | 41.65     |
> | **MeetingBank**|          |       | 48.06       | 18.61  | 15.37     | 11.71      | 13.21      | 19.76     |
> | **Py150**      |          |       |             | 57.84  | 19.11     | 44.98      | 45.72      | 39.02     |
> | **ScienceQA**  |          |       |             |        | 69.8      | 54.7       | 52.5       | 57.71     |
> | **NumGLUE-cm** |          |       |             |        |           | 33.33      | 31.2       | 30.15     |
> | **NumGLUE-ds** |          |       |             |        |           |            | 53.85      | 42.15     |
> | **20Minuten**  |          |       |             |        |           |            |            | 40.88     |
>
> **References**
>
> [1] HFT: Half Fine-Tuning for Large Language Models
>
> [2] TRACE: A Comprehensive Benchmark for Continual Learning in Large Language Models
>
> ---
> We hope we have cleared your concerns about our work. We have also revised our manuscript according to your comments, and we would appreciate it if we can get your further feedback at your earliest convenience.

---

> ### Author Response · Authors · 2024-11-22
> **Any Further Suggestions from Reviewer UiHH on Our Reply to Your Previous Questions?**
>
> We would like to thank the reviewer again for your thoughtful reviews and valuable feedback.
>
> We would appreciate it if you could let us know if our responses have addressed your concerns and whether you still have any other questions about our rebuttal and answers to your previous questions.
>
> We would be happy to do any follow-up discussion or address any additional comments.

---

### Official Review · Reviewer_WWeU · 2024-11-02

**Soundness:** 3
**Presentation:** 2
**Contribution:** 2
**Rating:** 6
**Confidence:** 4

**Summary:**

This paper introduced a Mixture of Layer Experts (MoLEx), which leverages layers as experts to facilitate the exchange of linguistic information and improve a model’s fine-tuning and transfer knowledge ability.
In particular, the author proposed a structural change to the architecture of the model which is compatible with PEFT method while maintaining the same number of effective parameters.
Experimental results demonstrated that MoLEx significantly improves performance across a range of downstream tasks, including the GLUE benchmark and the E2E Challenge, while incurring minimal additional computational overhead and scales well with model size.

**Strengths:**

I think this paper is a good upgrade in the field of Parameter Efficient Fine-Tuning.
Comparing with predominant method (Lora),  it achieve significant improvement on a few data-set with almost the same cost (speed, memory, parameter size).
Also I like the theoretical proof to justify the robustness of MoLEx in a simplified model and the empirical evidence provided.

**Weaknesses:**

1.My major concern of this paper is the linguistic analysis part.  One of the main claim of this paper is "relevant linguistic information is captured by selected experts".  I think it's very intuitive and hard to explain. In Figure1, the author tried to connect some linguistic feature with different layers in a probing task. But I am not quite convinced by the analysis, I think most NLP task involves different level of linguistic information, so it's hard to say one task focus more on certain part of information and the proposed method could facilitate the process.  So would like to see more discussion on this direction or if there is another angle to explain where the performance gain is coming from.

2.I feel the real computational cost of the proposed model is larger than Lora, but the parallelism makes the final speed the same. Is there any statics on this issue?

3.One small issue is the presentation of Lemma and Theorem looks a little messy, the alignment and layout could be adjusted to make it easy to read.

**Questions:**

please see weakness part

---

> ### Author Response · Authors · 2024-11-19
> **Review Reply (1/2): Q1**
>
> **Q1. My major concern of this paper is the linguistic analysis part. One of the main claim of this paper is "relevant linguistic information is captured by selected experts". I think it's very intuitive and hard to explain. In Figure1, the author tried to connect some linguistic feature with different layers in a probing task. But I am not quite convinced by the analysis, I think most NLP task involves different level of linguistic information, so it's hard to say one task focus more on certain part of information and the proposed method could facilitate the process. So would like to see more discussion on this direction or if there is another angle to explain where the performance gain is coming from.**
>
> **Answer:** Thank you for your question. We agree with the reviewer that the claim of "relevant linguistic information is captured by selected experts" is intuitive but might be difficult to understand. We will provide a more detailed explanation of how the claim can be empirically validated using the probing tasks as in seen Table 4 of our manuscript and its connection with NLP tasks.
>
> Each probing task is a classification task consisting of sentences and labels that represent a certain type of linguistic information. For example, BShift is a binary classification task where half the dataset has sentences intact and another half has sentences with 2 random adjacent words inverted. The goal is to predict if the sentence has a legal word order or if it has been inverted, which is a form of syntatic information. We input each sentence in the dataset into a language model and extract out the hidden representations, the outputs, of the sentence in each layer. Then, a classifier is trained on these representations and their labels. A high classifier score indicates that the representations at that layer effectively encodes the type of linguistic information that the dataset represents. This is a standard analysis performed to understand the linguistic properties in a model as proposed in [1].
>
> This is the primary reason for bolding the top 2 results of each probe task in Table 4 of our manuscript, to highlight the type of linguistic information captured by each layer of the model. From our results, in a RoBERTa model, the early layers contain superficial information, the middle layers, syntactic information and later layers, semantic information.
>
> Now that we have established the linguistic feature in each layer, we can form the connection between that and our proposed method. We agree with the reviewer that each NLP task should involve different levels of linguistic information. That is the reason we believe our method works well, as a broader variety of information is provided to the model at each layer by combining layers as experts. At each layer in MoLEx, we combine different experts, each capturing distinct linguistic information, to provide the model with a more comprehensive understanding of the input data to make better predictions. Hence, generally, we see in Figure 1 of the paper, most of the chosen experts include layers that represent different types of linguistic information i.e. on the off-diagonal blocks.
>
> However, there are certain tasks that require only specific types of information about the sentence to complete the task. These are the tasks that should prioritize choosing layers that encode for the information needed. CoLA is one example as it is a classification task on grammatical acceptibility. To solve the task, the model should mainly require information about grammer like tenses and noun or verb placements. These are captured by the later layers and as seen in the heatmap for CoLA in Figure 4 (left subplot), these are the layer blocks chosen the most frequently (last column of the grid).
>
> **References**
>
> [1] Conneau, Alexis, German Kruszewski, Guillaume Lample, Loïc Barrault, and Marco Baroni. "What you can cram into a single vector: Probing sentence embeddings for linguistic properties." arXiv preprint arXiv:1805.01070 (2018).

---

> ### Author Response · Authors · 2024-11-19
> **Review Reply (2/2): Q2, Q3**
>
> **Q2. I feel the real computational cost of the proposed model is larger than Lora, but the parallelism makes the final speed the same. Is there any statics on this issue?**
>
> **Answer:** We agree with the reviewer that real computational cost of the proposed model is larger than the naive PEFT method, but the parallelism makes the final speed the same.
>
> While MoLEx almost doubles the overall computational load (flops), there is only a minimal increase in inference time due to parallelization of the forward computations through two layers. We provide a more detailed efficiency analysis in the Table 1 below for a more accurate comparison with baseline models and have included this in Appendix E.3 of our revised manuscript.
>
> Table 1: Efficiency analysis of Llama-3.2-1B fine-tuned using LoRA during inference on the Alpaca dataset with and without MoLEx implemented.
>
> | Model \ Metric | Total Parameters | Trainable Parameters| Trainable Parameters (%) |Memory (MB)| Flop/Sample  | Sec/Sample | Flop/Sec |
> | -------- | -------- | -------- |-------- |-------- |-------- |-------- |-------- |
> | Llama-3.2-1B (Baseline) | 1,236,666,368  | 851,968  |0.0689  |10,442     | 12.329 T | 0.506 | 24.366 T |
> | Llama-3.2-1B (MoLEx)  | 1,236,699,152  | 884,752     |0.0715     |10,442     |22.511 T | 0.557 | 40.415 T |
>
> As we see in Table 1, while the number of flops per sample is almost doubled, the amount of time per sample is only slightly increased, maintaining the efficiency of the model in terms of run time per computational load.
>
> **Q3. One small issue is the presentation of Lemma and Theorem looks a little messy, the alignment and layout could be adjusted to make it easy to read.**
>
> **Answer:** Thank you for your comments. We have centered the assumptions in Theorem 1 and prevent line breaks on equations in the revised manuscript. We would greatly appreciate any additional suggestions you could provide on improving the alignment and layout of the Lemma and Theorem in our paper to enhance readability. We will incorporate your recommendations to refine the presentation of the Lemma and Theorem in our final revision.
>
> -----
> We hope we have cleared your concerns about our work. We have also revised our manuscript according to your comments, and we would appreciate it if we can get your further feedback at your earliest convenience.

---

> > ### Comment · Reviewer_WWeU · 2024-11-22
> >
> > Thanks for addressing some of my concerns. For Q2 and Q3, glad to see the updates on the paper, it will make the paper better.
> > For Q1, still not quite convincing, I think the author agree with my point for the most part. I will raise my score accordingly.

---

> > > ### Author Response · Authors · 2024-11-22
> > > **Thanks for your endorsement!**
> > >
> > > Thanks for your response, and we appreciate your endorsement.

---

### Official Review · Reviewer_D753 · 2024-11-02

**Soundness:** 3
**Presentation:** 3
**Contribution:** 3
**Rating:** 8
**Confidence:** 3

**Summary:**

This paper introduces a new approach to fine-tune LLMs. PEFT is becoming a mainstream method to adapt general-purpose model to specific tasks.MoLEx, proposed by the authors, aims at improving PEFT through leveraging pre-trained model layers and MoE to provide the model with more structural knowledge about the data. This method enhances information exchange between layers, leading to better predictions with minimal computational overhead. Experiments show that MoLEx is a more efficient technique to fine tune models.

**Strengths:**

1. The paper has clearly explained the benefits of using MoLEx to sparse upcycling models and its robustness as an ensemble model. Theoretical proof has laid good foundation for the idea.
2. The idea of exploiting MoE to improve the efficiency of LLM fine-tuning is a novel idea and a problem worth to research.
3. The experiments are well designed and show comprehensive results on the proposed idea. Although I do think it is necessary to further expand to more models and tasks, the promising results are a good start.

**Weaknesses:**

1. I am a bit confused by the inference time. I can understand MoE inference can be accelerated with parallel/distributed computation. But it also means using more resources. It is better to clarify what is the fair comparison in terms of the efficiency with naive PEFT models.

**Questions:**

1. How to determine K in top-K and the number of experts  in the upcycling process. I know Top-1 is widely used number. But I am not sure in this paper, how will it impact the performance.

---

> ### Author Response · Authors · 2024-11-19
> **Review Reply: Q1, Q2**
>
> **Q1. I am a bit confused by the inference time. I can understand MoE inference can be accelerated with parallel/distributed computation. But it also means using more resources. It is better to clarify what is the fair comparison in terms of the efficiency with naive PEFT models.**
>
> **Answer:** We agree with the reviewer that while more resources are required during MoE inference, these can be accelerated with parallel/distributed computation.
>
> As computational efficiency refers to the amount of time required for a given step in a calculation, we maintain that MoLEx is as efficient as the original method used without MoLEx. While MoLEx almost doubles the overall computational load (flops), there is only a minimal increase in inference time due to parallelization of the forward computations through two layers. We provide a more detailed efficiency analysis in the Table 1 below for a more accurate comparison with naive PEFT models. We also include our detailed explanation in Appendix E.3 of our revised manuscript.
>
> Table 1: Efficiency analysis of Llama-3.2-1B fine-tuned using LoRA during inference on the Alpaca dataset with and without MoLEx implemented.
>
> | Model \ Metric | Total Parameters | Trainable Parameters| Trainable Parameters (%) |Memory (MB)| Flop/Sample  | Sec/Sample | Flop/Sec |
> | -------- | -------- | -------- |-------- |-------- |-------- |-------- |-------- |
> | Llama-3.2-1B (Baseline) | 1,236,666,368  | 851,968  |0.0689  |10,442     | 12.329 T | 0.506 | 24.366 T |
> | Llama-3.2-1B (MoLEx)  | 1,236,699,152  | 884,752     |0.0715     |10,442     |22.511 T | 0.557 | 40.415 T |
>
> As we see in Table 1, while the number of flops per sample is almost doubled, the amount of time per sample is only slightly increased, maintaining the efficiency of the model in terms of run time per computational load.
>
> **Q2. How to determine K in top-K and the number of experts in the upcycling process. I know Top-1 is widely used number. But I am not sure in this paper, how will it impact the performance.**
>
> **Answer:** Thanks for your comments. As mentioned in Section 4.3 of our manuscript, in Table 9 in Appendix D, we evaluate 3 GLUE tasks—CoLA, QQP, and SST-2—using both Top-1 and Top-2 routing. Our results indicate that Top-1 routing consistently performs better. Therefore, we adopt Top-1 routing for MoLEx. For convenience, we display our results below as well.
>
> Table 9: Comparison of RoBERTa-base on GLUE tasks, CoLA, QQP and SST-2 when fine-tuned with MoLEx using Top-1 and Top-2 routing. We report accuracy for all tasks in the table below.
>
> |Method |  CoLA | QQP | SST-2 |
> |--|--|--|--|
> |MoLEx (Top1) |**64.8$\pm$.5** | **91.0$\pm$.0** | **95.4$\pm$.2**  |
> |MoLEx  (Top2)| 63.7$\pm$.4 | 90.7$\pm$.0 | 95.0$\pm$0.3|
>
> -----
> We hope we have cleared your concerns about our work. We have also revised our manuscript according to your comments, and we would appreciate it if we can get your further feedback at your earliest convenience.

---

> ### Author Response · Authors · 2024-11-22
> **Any Questions from Reviewer D753 on Our Rebuttal?**
>
> We would like to thank the reviewer again for your thoughtful reviews and valuable feedback.
>
> We would appreciate it if you could let us know if our responses have addressed your concerns and whether you still have any other questions about our rebuttal.
>
> We would be happy to do any follow-up discussion or address any additional comments.

---

> > ### Comment · Reviewer_D753 · 2024-11-27
> >
> > Thank you very much for the response. It addresses my concerns.

---

> > > ### Author Response · Authors · 2024-11-27
> > > **Thanks for your endorsement!**
> > >
> > > Thanks for your response, and we appreciate your endorsement.

---

### Author Response · Authors · 2024-11-19
**Global Reply**

Dear AC and reviewers,

Thanks for your thoughtful reviews and valuable comments, which have helped us improve the paper significantly. We are encouraged by the endorsements that: 1) (Reviewers D753, WWeU, UiHH) MoLEx is a novel, simple and effective method to upgrade paramter efficient fine-tuning methods, 2) (Reviewers D753, WWeU) the theoretical proof in a simplified model has laid good foundation for the idea and to justify its robustness, and 3) (Reviewers D753, WWeU, UiHH) comprehensive experiments have demonstrated that MoLEx has significant performance gains with almost the same computational speed and memory.

One of the main concerns from the reviewers is to provide a more detailed and fair comparison with naive PEFT models as MoLEx does increase computational cost but maintains its computational efficiency through parallelism. Another concern is to expand our experiments to include more recent LLMs, such as Qwen-2.5 or Llama-3.1, and other fine-tuning methods, to enhance the credibility of our findings. We address these questions here and have updated our submission based on the reviewers' feedback. Our revisions are highlighted in blue.

**Q1: Detailed efficiency analysis of MoLEx and naive PEFT models**

**Answer:** We agree with the reviewers that real computational cost of the proposed model is larger than the naive PEFT method, but the parallelism makes the final speed the same.

While MoLEx almost doubles the overall computational load (flops), there is only a minimal increase in inference time due to parallelization of the forward computations through two layers. We provide a more detailed efficiency analysis in the Table 1 below for a more accurate comparison with baseline models and have included this in Appendix E.3 of our revised manuscript.

Table 1: Efficiency analysis of Llama-3.2-1B fine-tuned using LoRA during inference on the Alpaca dataset with and without MoLEx implemented.

| Model \ Metric | Total Parameters | Trainable Parameters| Trainable Parameters (%) |Memory (MB)| Flop/Sample  | Sec/Sample | Flop/Sec |
| -------- | -------- | -------- |-------- |-------- |-------- |-------- |-------- |
| Llama-3.2-1B (Baseline) | 1,236,666,368  | 851,968  |0.0689  |10,442     | 12.329 T | 0.506 | 24.366 T |
| Llama-3.2-1B (MoLEx)  | 1,236,699,152  | 884,752     |0.0715     |10,442     |22.511 T | 0.557 | 40.415 T |

As we see in Table 1, while the number of flops per sample is almost doubled, the amount of time per sample is only slightly increased, maintaining the efficiency of the model in terms of run time per computational load.

**Q2: Experiments on more recent LLMs (Llama-3.2-1B) and other fine-tuning methods (full parameter fine-tuning)**

**Answer:** We have conducted additional experiments to fine-tune Llama-3.2-1B on the Alpaca dataset using LoRA. We use the publicly available repository https://github.com/meta-llama/llama3 for our experiments and employ MoLEx to fine-tune the model in comparison with LoRA. As shown in Table 2 below, on this task with Llama-3.2-1B, MoLEx achieves better train and validation PPL than LoRA. These results further justify the effectiveness of our method and demonstrate our compatibility with recent LLMs. We have also included these results in Appendix E.2, Table 11, in our revision.

Table 2: Train and validation perplexity (PPL) when fine-tuning Llama-3.2-1B on Alpaca using LoRA and LoRA + MoLEx. Lower PPL is indicative of better performance.

| Model \ Metric| Train PPL ($\downarrow$) | Validation PPL ($\downarrow$) |
| -------- | -------- | -------- |
| LoRA  | 4.18    | 4.11     |
| MoLEx  | **4.05**    | **4.02**   |

Further, as Reviewer UiHH noted, MoLEx is adaptable to various models and training methods. Following the reviewer's suggestion, we have conducted additional experiments for RoBERTa using full parameter fine-tuning on 4 GLUE tasks. We present the results in Table 3 below. MoLEx consistently outperforms the full parameter fine-tuning across all tasks. These findings further confirm MoLEx's adaptability to different models and training methods. We have also included these results in Appendix E.1, Table 10, in our revision.

Table 3: RoBERTa with full parameter fine-tuning and with MoLEx when fine-tuned on 4 GLUE tasks, RTE, MRPC, STS-B and CoLA. We report accuracy for RTE and MRPC, Pearson correlation for STS-B and Matthew's correlation for CoLA. A higher value reflects a better performance of the model.

| Model \ Task| RTE | MRPC | STS-B |CoLA |
| -------- | -------- | -------- |--|--|
| RoBERTa (full parameter)  | 80.1    | 88.7     | 90.9 |62.6|
| RoBERTa (MoLEx)  | **80.9**    | **89.5**   | **91.1**|**63.3**|


-----

We are glad to answer any further questions you have on our submission.

---

### Author Response · Authors · 2024-11-19
**Summary of revision:**

Incorporating the insightful comments from the reviewers into our paper has helped us improve its quality. Here, we provide a summary of the revisions (highlighted in blue in the revised manuscript) for convenience:

1. Section 2.2: We clarify the implementation of MoLEx with parameter-efficient fine-tuning methods to avoid confusion. A diagram is also included in Figure 1 for easy visualization of our method.
2. Section 2.3: We fix some alignment issues and equation line breaks in  Theorem 1, Lemma 1 and Corollary 1 and 2 for an easier read and neater presentation.
3. Section 3.1, Table 1; Section 3.2, Table 2: We make the total number of trainable parameters in MoLEx more exact to reflect the slight increase in the parameter count due to an additional implementation of a gating function.
4. Appendix E.1: We provide additional experimental results on full parameter fine-tuning for RoBERTa on GLUE to demonstrate that MoLEx is adaptable to various training methods, besides LoRA.
5. Appendix E.2: We provide additional experimental results on fine-tuning Llama-3.2-1B on the Alpaca dataset to illustrate that MoLEx is compatible with recent large language models. We also evaluate the baseline naive LoRA model and MoLEx on the standard benchmark MMLU for further comparison.
6. Appendix E.3: We include a more detailed efficiency analysis of MoLEx with naive parameter-efficient fine-tuning models for a fair comparison between them.

---

### Author Response · Authors · 2024-11-22
**Additional Experimental Results**

Dear AC and reviewers,

We would like to thank all reviewers again for their thoughtful reviews and valuable feedback. We have obtained additional results on GLUE and MMLU and added more statistics for our efficiency analysis. We summarize these results below:

(1) As the reviewer UiHH noted, MoLEx is adaptable to various models and training methods. Following the reviewer's suggestion, we conducted  experiments for RoBERTa using full parameter fine-tuning on 4 GLUE tasks, RTE, MRPC, STS-B and CoLA.

We have finished running additional results for this experiment on the remaining GLUE tasks and report our full results below in Table 1. We include results on the previous tasks reported as well for convenience. The same results can be found in Table 10 in Appendix E.3 of our revised manuscript. Our MoLEx consistently surpasses full parameter fine-tuning on each task, highlighting its versatility across various models and training methods.

Table 1: RoBERTa with full parameter fine-tuning and with MoLEx when fine-tuned on the GLUE benchmark. We report accuracy for all tasks except for, Pearson correlation for STS-B, Matthew's correlation for CoLA and the overall (matched and mismatched) accuracy for MNLI. A higher value reflects a better performance of the model.
 | Model \ Task| RTE | MRPC | STS-B |CoLA |MNLI | QNLI | SST-2 |QQP |Ave.|
| -------- | -------- | -------- |--|--|--|--|--|--|--|
| RoBERTa (full parameter)  | 80.1    | 88.7     | 90.9 |62.6| 87.7|92.9 | 95.0|91.8 |86.2|
| RoBERTa (MoLEx)  | **80.9**    | **89.5**   | **91.1**|**63.3**|**87.8** | **93.1**|**95.1** | 91.8| **86.6**|

(2) From reviewer UiHH's feedback, PPL is generally not used to evaluate a model's performance during the fine-tuning stage. Instead, benchmarks like MMLU are typically used for evaluation. Following the reviewer's suggestion, we have also run the evaluation on MMLU using Llama-3.2-1B. We report our results in Table 2 below. We improved by about 1% on MMLU, further verifying the effectiveness of MoLEx. We also include these results in Table 12 in Appendix E.2 of our revision.

Table 2: Accuracy when evaluating Llama-3.2-1B on MMLU using LoRA and LoRA + MoLEx. A higher value is indicative of better performance.
| Model \ Metric| Acc. % ($\uparrow$) |
 | -------- | -------- |
| LoRA  | 30.42 |
| MoLEx  | **31.51**    |

(3) We add one more statistic, mins/epoch of training, to our detailed efficiency analysis on MoLEx and naive PEFT methods. Fine-tuning on the Alpaca dataset with Llama-3.2-1B using LoRA takes 4:59 minutes per epoch, whereas using MoLEx requires 6:00 minutes per epoch. We summarize these results in Table 3 below. Same results can be found in Table 13 in Appendix E.3 of our revised manuscript.

Table 3: Time in minutes when training Llama-3.2-1B on the Alpaca dataset using LoRA and LoRA + MoLEx for 1 epoch.
| Model \ Metric| min/epoch |
| -------- | -------- |
| LoRA  | 4:59 |
| MoLEx  | 6:00   |

----
We would be happy to do any follow-up discussion or address any additional comments.

---

### Meta-Review · Area_Chair_Q3UV · 2024-12-20

**Metareview:**

This paper introduces MoLEx (Mixture of Layer Experts) for parameter-efficient fine-tuning of LLMs. The key idea is to treat existing layers in pre-trained models as experts and employ a routing mechanism to selectively combine them during fine-tuning. The authors demonstrate improved performance across multiple benchmarks while maintaining efficiency through parallel computation.

The idea and the proposed approach is novel through repurposing the layers as experts (without introducing additional parameters). The empirical experiments are also quite comprehensive to support the major claim. Reviewers also raised some concerns regarding the computational efficiency claims, as well as limited exploration of token-level routing approaches, which authors need to consider later.

While there were some concerns regarding the presentation and computation overhead, the core contribution - repurposing existing layers as experts for improved fine-tuning - is novel and interesting, and might inspire future studies. Therefore, I'm inclined to accept this paper and recommend the authors carefully consider those suggestions to revise the paper's presentation logic for the final version.

**Additional Comments On Reviewer Discussion:**

While there were initial concerns, the authors adequately addressed them during the discussion period. It also underwent internal discussion following the public discussion phase, and I personally verified many of its details. Overall I believe the authors' rebuttal resolved most of the reviewers' concerns. The remaining issues primarily relate to presentation style and can be (and should be) addressed in the final version.

---

### Decision · Program_Chairs · 2025-01-22

Accept (Poster)